# A Convex Relaxation Barrier to Tight Robustness Verification of Neural Networks

**Hadi Salman**[*]
Microsoft Research AI
hadi.salman@microsoft.com

**Greg Yang**
Microsoft Research AI
gregyang@microsoft.com

**Huan Zhang**
UCLA
huan@huan-zhang.com

**Cho-Jui Hsieh**
UCLA
chohsieh@cs.ucla.edu

**Pengchuan Zhang**
Microsoft Research AI
penzhan@microsoft.com

## Abstract

Verification of neural networks enables us to gauge their robustness against adversarial attacks. Verification algorithms fall into two categories: *exact* verifiers that run in exponential time and *relaxed* verifiers that are efficient but incomplete. In this paper, we unify all existing LP-relaxed verifiers, to the best of our knowledge, under a general convex relaxation framework. This framework works for neural networks with diverse architectures and nonlinearities and covers both primal and dual views of neural network verification. Next, we perform large-scale experiments, amounting to more than 22 CPU-years, to obtain exact solution to the convex-relaxed problem that is optimal within our framework for ReLU networks. We find the exact solution does not significantly improve upon the gap between PGD and existing relaxed verifiers for various networks trained normally or robustly on MNIST and CIFAR datasets. Our results suggest there is an inherent *barrier* to tight verification for the large class of methods captured by our framework. We discuss possible causes of this barrier and potential future directions for bypassing it. Our code and trained models are available at http://github.com/Hadisalman/robust-verify-benchmark[2].

## 1 Introduction

A classification neural network $f : \mathbb{R}^n \to \mathbb{R}^K$ (where $f_i(x)$ should be thought of as the $i$th logit) is considered *adversarially robust* with respect to an input $x$ and its neighborhood $\mathcal{S}_{in}(x)$ if

$$\min_{x' \in \mathcal{S}_{in}(x), i \neq i^*} f_{i^*}(x) - f_i(x') > 0, \quad \text{where} \quad i^* = \arg\max_j f_j(x). \tag{1}$$

Many recent works have proposed robustness verification methods by lower-bounding eq. (1); the positivity of this lower bound proves the robustness w.r.t. $\mathcal{S}_{in}(x)$. A dominant approach thus far has tried to relax eq. (1) into a convex optimization problem, from either the primal view [Zhang et al., 2018, Gehr et al., 2018, Singh et al., 2018, Weng et al., 2018] or the dual view [Wong and Kolter, 2018, Dvijotham et al., 2018b, Wang et al., 2018b]. In **our first main contribution**, we propose a layer-wise convex relaxation framework that unifies these works and reveals the relationships between them (Fig. 1). We further show that the performance of methods within this framework is subject to a theoretical limit: the performance of the optimal layer-wise convex relaxation.

This then begs the question: is the road to fast and accurate robustness verification paved by just faster and more accurate layer-wise convex relaxation that approaches the theoretical limit? In our **second main contribution**, we answer this question in the *negative*. We perform extensive experiments

---

[*]Work done as part of the Microsoft AI Residency Program.
[2]Please see http://arxiv.org/abs/1902.08722 for the full and most recent version of this paper.

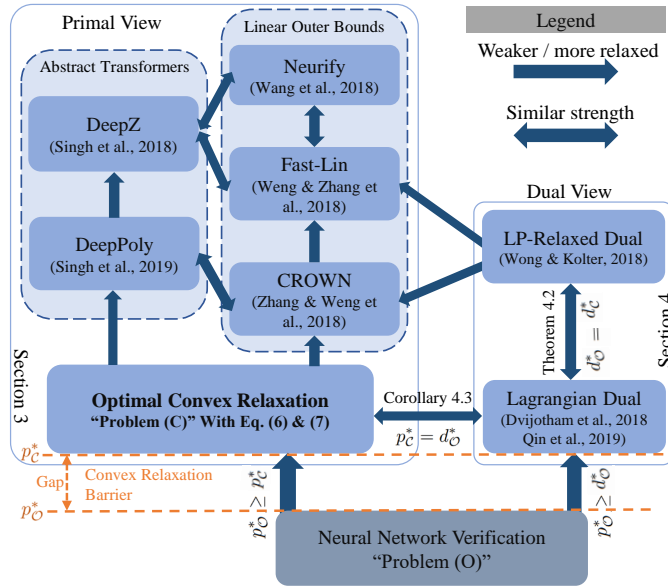

Figure 1: Relationship between existing relaxed algorithms and our framework. See Appendix D for detailed discussions of each unlabeled arrow from the "Primal view" side.

with deep ReLU networks to compute the optimal layer-wise convex relaxation and compare with the LP-relaxed dual formulation from Wong and Kolter [2018], the PGD attack from Madry et al. [2017], and the mixed integer linear programming (MILP) exact verifier from Tjeng et al. [2019]. Over different models, sizes, training methods, and datasets (MNIST and CIFAR-10), we find that (i) in terms of lower bounding the *minimum $l_\infty$ adversarial distortion*[3], the optimal layer-wise convex relaxation only slightly improves the lower bound found by Wong and Kolter [2018], especially when compared with the upper bound provided by the PGD attack, which is consistently 1.5 to 5 times larger; (ii) in terms of upper bounding the *robust error*, the optimal layer-wise convex relaxation does not significantly close the gap between the PGD lower bound (or MILP exact answer) and the upper bound from Wong and Kolter [2018]. Therefore, there seems to be an inherent barrier blocking our progress on this road of layer-wise convex relaxation, and we hope this work provokes much thought in the community on how to bypass it.

## 2 Preliminaries and Related Work

**Exact verifiers and NP-completeness.** For ReLU networks (piece-wise linear networks in general), exact verifiers solve the robustness verification problem (1) by typically employing MILP solvers [Cheng et al., 2017, Lomuscio and Maganti, 2017, Dutta et al., 2018, Fischetti and Jo, 2017, Tjeng et al., 2019, Xiao et al., 2019] or Satisfiability Modulo Theories (SMT) solvers [Scheibler et al., 2015, Katz et al., 2017, Carlini et al., 2017, Ehlers, 2017]. However, due to the NP-completeness for solving such a problem [Katz et al., 2017, Weng et al., 2018], it can be really challenging to scale these to large networks. It can take Reluplex [Katz et al., 2017] several hours to find the minimum distortion of an example for a ReLU network with 5 inputs, 5 outputs, and 300 neurons. A recent work by Tjeng et al. [2019] uses MILP to exactly verify medium-size networks, but the verification time is very sensitive to how a network is trained; for example, it is fast for networks trained using the LP-relaxed dual formulation of Wong and Kolter [2018], but much slower for normally trained networks. A concurrent work by Xiao et al. [2019] trains networks with the objective of speeding up the MILP verification problem, but this compromises on the performance of the network.

**Relaxed and efficient verifiers.** These verifiers solve a relaxed, but more computationally efficient, version of (1), and have been proposed from different perspectives. From the primal view, one can relax the nonlinearity in (1) into linear inequality constraints. This perspective has been previously explored as in the framework of "abstract transformers" [Singh et al., 2018, 2019a,b, Gehr et al.,

2018, Mirman et al., 2018], via linear outer bounds of activation functions [Zhang et al., 2018, Weng et al., 2018, Wang et al., 2018a,b], or via interval bound propagation [Gowal et al., 2018, Mirman et al., 2018]. From the dual view, one can study the dual of the relaxed problem [Wong and Kolter, 2018, Wong et al., 2018] or study the dual of the original nonconvex verification problem [Dvijotham et al., 2018b,a, Qin et al., 2019]. In this paper, we unify both views in a common convex relaxation framework for NN verification, clarifying their relationships (as summarized in Fig. 1).

Raghunathan et al. [2018b] formulates the verification of ReLU networks as a quadratic programming problem and then relaxes and solves this problem with a semidefinite programming (SDP) solver. While our framework does not cover this SDP relaxation, it is not clear to us how to extend the SDP relaxed verifier to general nonlinearities, for example max-pooling, which can be done in our framework on the other hand. Other verifiers have been proposed to certify via an intermediary step of bounding the local Lipschitz constant [Hein and Andriushchenko, 2017, Weng et al., 2018, Raghunathan et al., 2018a, Zhang et al., 2019], and others have used *randomized smoothing* to certify with high-probability [Lecuyer et al., 2018, Li et al., 2018, Cohen et al., 2019, Salman et al., 2019]. These are outside the scope of our framework.

Combining exact and relaxed verifiers, hybrid methods have shown some effectiveness [Bunel et al., 2018, Singh et al., 2019b]. In fact, many exact verifiers also use relaxation as a subroutine to speed things up, and hence can be viewed as hybrid methods as well. In this paper, we are not concerned with such techniques but only focus on *relaxed verifiers*.

## 3 Convex Relaxation from the Primal View

**Problem setting.** In this paper, we assume that the neighborhood $\mathcal{S}_{in}(x^{\text{nom}})$ is a convex set. An example of this is $\mathcal{S}_{in}(x^{\text{nom}}) = \{x : \|x - x^{\text{nom}}\|_\infty \leq \epsilon\}$, which is the constraint on $x$ in the $\ell_\infty$ adversarial attack model. We also assume that $f(x)$ is an $L$-layer feedforward NN. For notational simplicity, we denote $\{0, 1, \ldots, L-1\}$ by $[L]$ and $\{x^{(0)}, x^{(1)}, \ldots, x^{(L-1)}\}$ by $x^{[L]}$. We define $f(x)$ as,

$$x^{(l+1)} = \sigma^{(l)}(\mathbf{W}^{(l)}x^{(l)} + b^{(l)}) \quad \forall l \in [L], \quad \text{and} \quad f(x) := z^{(L)} = \mathbf{W}^{(L)}x^{(L)} + b^{(L)}, \quad (2)$$

where $x^{(l)} \in \mathbb{R}^{n^{(l)}}$, $z^{(l)} \in \mathbb{R}^{n_z^{(l)}}$, $x^{(0)} := x \in \mathbb{R}^{n^{(0)}}$ is the input, $\mathbf{W}^{(l)} \in \mathbb{R}^{n_z^{(l)} \times n^{(l)}}$ and $b^{(l)} \in \mathbb{R}^{n_z^{(l)}}$ are the weight matrix and bias vector of the $l^{\text{th}}$ linear layer, and $\sigma^{(l)} : \mathbb{R}^{n_z^{(l)}} \to \mathbb{R}^{n^{(l+1)}}$ is a (nonlinear) activation function like (leaky-)ReLU, the sigmoid family (including sigmoid, arctan, hyperbolic tangent, etc), and the pooling family (MaxPool, AvgPool, etc). Our results can be easily extended to networks with convolutional layers and skip connections as well, similar to what is done in Wong et al. [2018], as these can be seen as special forms of (2).

Consider the following optimization problem $\mathcal{O}(c, c_0, L, \underline{z}^{[L]}, \overline{z}^{[L]})$:

$$\begin{aligned} \min_{(x^{[L+1]}, z^{[L]}) \in \mathcal{D}} & \quad c^\top x^{(L)} + c_0 \\ \text{s.t.} & \quad z^{(l)} = \mathbf{W}^{(l)}x^{(l)} + b^{(l)}, l \in [L], \\ & \quad x^{(l+1)} = \sigma^{(l)}(z^{(l)}), l \in [L], \end{aligned} \quad (\mathcal{O})$$

where the optimization domain $\mathcal{D}$ is the set of activations and preactivations $\{x^{(0)}, x^{(1)}, \ldots, x^{(L)}, z^{(0)}, z^{(1)}, \ldots, z^{(L-1)}\}$ satisfying the bounds $\underline{z}^{(l)} \leq z^{(l)} \leq \overline{z}^{(l)} \; \forall l \in [L]$, i.e.,

$$\mathcal{D} = \left\{ (x^{[L+1]}, z^{[L]}) : x^{(0)} \in \mathcal{S}_{in}(x^{\text{nom}}), \quad \underline{z}^{(l)} \leq z^{(l)} \leq \overline{z}^{(l)}, l \in [L] \right\}. \quad (3)$$

If $c^\top = \mathbf{W}_{i^{\text{nom}},:}^{(L)} - \mathbf{W}_{i,:}^{(L)}$, $c_0 = b_{i^{\text{nom}}}^{(L)} - b_i^{(L)}$, $\underline{z}^{[L]} = -\infty$, and $\overline{z}^{[L]} = \infty$, then ($\mathcal{O}$) is equivalent to problem (1). However, when we have better information about valid bounds $\underline{z}^{[l]}$ and $\overline{z}^{[l]}$ of $z^{[l]}$, we can significantly narrow down the optimization domain and, as will be detailed shortly, achieve tighter solutions when we relax the nonlinearities. We denote the minimal value of $\mathcal{O}(c, c_0, L, \underline{z}^{[L]}, \overline{z}^{[L]})$ by $p^*(c, c_0, L, \underline{z}^{[L]}, \overline{z}^{[L]})$, or just $p_\mathcal{O}^*$ when no confusion arises.

**Obtaining lower and upper bounds $(\underline{z}^{[L]}, \overline{z}^{[L]})$ by solving sub-problems.** This can be done by *recursively* solving ($\mathcal{O}$) with specific choices of $c$ and $c_0$, which is a common technique used in many

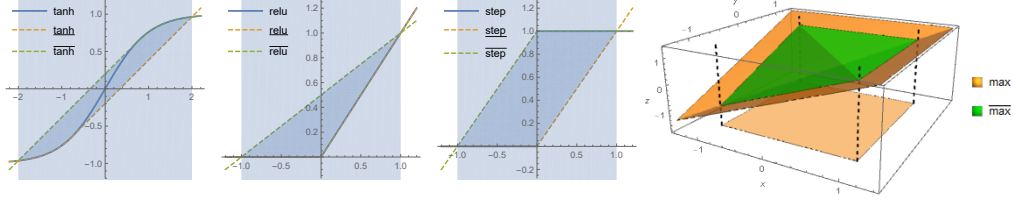

Figure 2: Optimal convex relaxations for common nonlinearities. For $\tanh$, the relaxation contains two linear segments and parts of the $\tanh$ function. For ReLU and the step function, the optimal relaxations are written as 3 and 4 linear constraints, respectively. For $z = \max(x, y)$, the light orange shadow indicates the pre-activation bounds for $x$ and $y$, and the optimal convex relaxation is lower bounded by the $\max$ function itself.

works [Wong and Kolter, 2018, Dvijotham et al., 2018b]. For example, one can obtain $\underline{z}_j^{(\ell)}$, a lower bound of $z_j^{(\ell)}$, by solving $\mathcal{O}(\mathbf{W}_{j,:}^{(\ell)\top}, b_j^{(\ell)}, \ell, \underline{z}^{[\ell]}, \overline{z}^{[\ell]})$; this shows that one can estimate $\underline{z}^{(l)}$ and $\overline{z}^{(l)}$ inductively in $l$. However, we may have millions of sub-problems to solve because practical networks can have millions of neurons. Therefore, it is crucial to have *efficient* algorithms to solve $(\mathcal{O})$.

**Convex relaxation in the primal space.** Due to the nonlinear activation functions $\sigma^{(l)}$, the feasible set of $(\mathcal{O})$ is nonconvex, which leads to the NP-completeness of the neural network verification problem [Katz et al., 2017, Weng et al., 2018]. One natural idea is to do convex relaxation of its feasible set. Specifically, one can relax the nonconvex equality constraint $x^{(l+1)} = \sigma^{(l)}(z^{(l)})$ to convex inequality constraints, i.e.,

$$\min_{(x^{[L+1]}, z^{[L]}) \in \mathcal{D}} c^\top x^{(L)} + c_0 \quad \text{s.t.} \quad z^{(l)} = \mathbf{W}^{(l)} x^{(l)} + b^{(l)}, \underline{\sigma}^{(l)}(z^{(l)}) \leq x^{(l+1)} \leq \overline{\sigma}^{(l)}(z^{(l)}), \forall l \in [L], \quad (\mathcal{C})$$

where $\underline{\sigma}^{(l)}(z)$ ($\overline{\sigma}^{(l)}(z)$) is convex (concave) and satisfies $\underline{\sigma}^{(l)}(z) \leq \sigma^{(l)}(z) \leq \overline{\sigma}^{(l)}(z)$ for $\underline{z}^{(l)} \leq z \leq \overline{z}^{(l)}$. We denote the feasible set of $(\mathcal{C})$ by $\mathcal{S}_\mathcal{C}$ and its minimum by $p_\mathcal{C}^*$. Naturally, we have that $\mathcal{S}_\mathcal{C}$ is convex and $p_\mathcal{C}^* \leq p_\mathcal{O}^*$. For example, Ehlers [2017] proposed the following relaxations for the ReLU function $\sigma_{ReLU}(z) = \max(0, z)$ and MaxPool $\sigma_{MP}(z) = \max_k z_k$:

$$\underline{\sigma}_{ReLU}(z) = \max(0, z), \quad \overline{\sigma}_{ReLU}(z) = \frac{\overline{z}}{\overline{z} - \underline{z}}(z - \underline{z}), \tag{4}$$

$$\underline{\sigma}_{MP}(z) = \max_k z_k \geq \sum_k (z_k - \overline{z}_k) + \max_k \overline{z}_k, \quad \overline{\sigma}_{MP}(z) = \sum_k (z_k + \underline{z}_k) - \max_k \underline{z}_k. \tag{5}$$

**The optimal layer-wise convex relaxation.** As a special case, we consider the optimal layer-wise convex relaxation, where

$$\underline{\sigma}_{\text{opt}}(z) \text{ is the greatest convex function majored by } \sigma,$$
$$\overline{\sigma}_{\text{opt}}(z) \text{ is the smallest concave function majoring } \sigma. \tag{6}$$

A precise definition can be found in (12) in Appendix B. In Fig. 2, we show the optimal convex relaxation for several common activation functions. It is easy to see that (4) is the optimal convex relaxation for ReLU, but (5) is not optimal for the MaxPool function. Under mild assumptions (non-interactivity as defined in definition B.2), the optimal convex relaxation of a nonlinear layer $x = \sigma(z)$, i.e., its convex hull, is simply $\underline{\sigma}_{\text{opt}}(z) \leq x \leq \overline{\sigma}_{\text{opt}}(z)$ (see proposition B.3). We denote the corresponding optimal relaxed problem as $\mathcal{C}_{\text{opt}}$, with its objective $p_{\mathcal{C}_{\text{opt}}}^*$.

We emphasize that by *optimal*, we mean the optimal convex relaxation of the *single* nonlinear constraint $x^{(l+1)} = \sigma^{(l)}(z^{(l)})$ (see Proposition (B.3)) instead of the optimal convex relaxation of the nonconvex feasible set of the original problem $(\mathcal{O})$. As such, techniques as in [Anderson et al., 2018, Raghunathan et al., 2018b] are outside our framework; see appendix C for more discussions.

**Greedily solving the primal with linear bounds.** As another special case, when there are *exactly one* linear upper bound and one linear lower bound for each nonlinear layer in $(\mathcal{C})$ as follows:

$$\overline{\sigma}^{(l)}(z^{(l)}) := \overline{a}^{(l)} z^{(l)} + \overline{b}^{(l)}, \qquad \underline{\sigma}^{(l)}(z^{(l)}) := \underline{a}^{(l)} z^{(l)} + \underline{b}^{(l)}. \tag{7}$$

the objective $p_\mathcal{C}^*$ can be *greedily* bounded in a layer-by-layer manner. We can derive one linear upper and one linear lower bound of $z^L := c^T x^L + c_0$ with respect to $z^{(L-1)}$, using the fact that

$z^{(L)} = c^T\sigma^{(L-1)}(z^{(L-1)}) + c_0$ and that $\sigma^{(L-1)}(z^{(L-1)})$ is linearly upper and lower bounded by $\overline{\sigma}^{(L-1)}(z^{(L-1)})$ and $\underline{\sigma}^{(L-1)}(z^{(L-1)})$. Because a linear combination of linear bounds (coefficients are related to the entries in $c$) can be relaxed to a single linear bound, we can apply this technique again and replace $z^{(L-1)}$ with its upper and lower bounds with respect to $z^{(L-2)}$, obtaining the bound for $z^{(L)}$ with respect to $z^{(L-2)}$. Applying this repeatedly eventually leads to linear lower and upper bounds of $z^{(L)}$ with respect to the input $x^{(0)} \in \mathcal{S}_{in}(x^{\mathrm{nom}})$.

This perspective covers Fast-Lin [Weng et al., 2018], DeepZ [Singh et al., 2018] and Neurify [Wang et al., 2018b], where the proposed linear lower bound has the same slope as the upper bound, i.e., $\underline{a}^{(l)} = \overline{a}^{(l)}$. The resulting shape is referred to as a *zonotope* in Gehr et al. [2018] and Singh et al. [2018]. In CROWN [Zhang et al., 2018] and DeepPoly [Singh et al., 2019a], this restriction is lifted and they can achieve better verification results than Fast-Lin and DeepZ. Fig. 1 summarizes the relationships between these algorithms. Importantly, each of these works has its own merits on solving the verification problem; our focus here is to give a unified view on how they perform convex relaxation of the original verification problem ($\mathcal{O}$) in our framework. See Appendix D for more discussions and other related algorithms.

# 4 Convex Relaxation from the Dual View

We now tackle the verification problem from the dual view and connect it to the primal view.

**Strong duality for the convex relaxed problem.** As in Wong and Kolter [2018], we introduce the dual variables for ($\mathcal{C}$) and write its Lagrangian dual as

$$
g_\mathcal{C}(\mu^{[L]}, \underline{\lambda}^{[L]}, \overline{\lambda}^{[L]}) := \min_{(x^{[L+1]}, z^{[L]}) \in \mathcal{D}} c^\top x^{(L)} + c_0 + \sum_{l=0}^{L-1} \mu^{(l)\top}(z^{(l)} - \mathbf{W}^{(l)}x^{(l)} - b^{(l)})
$$
$$
- \sum_{l=0}^{L-1} \underline{\lambda}^{(l)\top}(x^{(l+1)} - \underline{\sigma}^{(l)}(z^{(l)})) + \sum_{l=0}^{L-1} \overline{\lambda}^{(l)\top}(x^{(l+1)} - \overline{\sigma}^{(l)}(z^{(l)})).
$$

(8)

By weak duality [Boyd and Vandenberghe, 2004],

$$
d_\mathcal{C}^* := \max_{\mu^{[L]}, \underline{\lambda}^{[L]} \geq 0, \overline{\lambda}^{[L]} \geq 0} g_\mathcal{C}(\mu^{[L]}, \underline{\lambda}^{[L]}, \overline{\lambda}^{[L]}) \leq p_\mathcal{C}^*,
$$

(9)

but in fact we can show strong duality under mild conditions as well (note that the following result cannot be obtained by trivially applying Slater's condition; see appendix E and fig. 4).

**Theorem 4.1** ($p_\mathcal{C}^* = d_\mathcal{C}^*$). *Assume that both $\underline{\sigma}^{(l)}$ and $\overline{\sigma}^{(l)}$ have a finite Lipschitz constant in the domain $[\underline{z}^{(l)}, \overline{z}^{(l)}]$ for each $l \in [L]$. Then strong duality holds between ($\mathcal{C}$) and (9).*

**The optimal layer-wise dual relaxation.** Theorem 4.1 shows that taking the dual of the layer-wise convex relaxed problem ($\mathcal{C}$) cannot do better than the original relaxation. To obtain a tighter dual problem, one could directly study the Lagrangian dual of the original ($\mathcal{O}$),

$$
g_\mathcal{O}(\mu^{[L]}, \lambda^{[L]}) := \min_\mathcal{D} c^\top x^{(L)} + c_0 + \sum_{l=0}^{L-1} \mu^{(l)\top}(z^{(l)} - \mathbf{W}^{(l)}x^{(l)} - b^{(l)}) + \sum_{l=0}^{L-1} \lambda^{(l)\top}(x^{(l+1)} - \sigma^{(l)}(z^{(l)})),
$$

(10)

where the min is taken over $\{(x^{[L+1]}, z^{[L]}) \in \mathcal{D}\}$. This was first proposed in Dvijotham et al. [2018b]. Note, again, by weak duality,

$$
d_\mathcal{O}^* := \max_{\mu^{[L]}, \lambda^{[L]}} g_\mathcal{O}(\mu^{[L]}, \lambda^{[L]}) \leq p_\mathcal{O}^*,
$$

(11)

and $d_\mathcal{O}^*$ would seem to be strictly better than $d_\mathcal{C}^*$. Unfortunately, they turn out to be equivalent:

**Theorem 4.2** ($d_\mathcal{O}^* = d_{\mathcal{C}_{\mathrm{opt}}}^*$). *Assume that the nonlinear layer $\sigma^{(l)}$ is non-interactive (definition B.2) and the optimal layer-wise relaxation $\underline{\sigma}_{opt}^{(l)}$ and $\overline{\sigma}_{opt}^{(l)}$ are defined in (6). Then the lower bound $d_{\mathcal{C}_{opt}}^*$ provided by the dual of the optimal layer-wise convex-relaxed problem (9) and $d_\mathcal{O}^*$ provided by the dual of the original problem (11) are the same.*

The complete proof is in Appendix F [4]. Theorem 4.2 combined with the strong duality result of Theorem 4.1 implies that the primal relaxation ($\mathcal{C}$) and the two kinds of dual relaxations, (9) and (11), are all blocked by the same *barrier*. As concrete examples:

**Corollary 4.3** ($p^*_{\mathcal{C}_{\text{opt}}} = d^*_{\mathcal{O}}$)**.** *Suppose that the nonlinear activation functions $\sigma^{(l)}$ for all $l \in [L]$ are (for example) among the following: ReLU, step, ELU, sigmoid, tanh, polynomials and max pooling with disjoint windows. Assume that $\underline{\sigma}^{(l)}_{opt}$ and $\overline{\sigma}^{(l)}_{opt}$ are defined in (6), respectively. Then we have that the lower bound $p^*_{\mathcal{C}_{opt}}$ provided by the primal optimal layer-wise relaxation ($\mathcal{C}$) and $d^*_{\mathcal{O}}$ provided by the dual relaxation (11) are the same.*

**Greedily solving the dual with linear bounds.** When the relaxed bounds $\underline{\sigma}$ and $\overline{\sigma}$ are linear as defined in (7), the dual objective (9) can be lower bounded as below:

$$p^*_{\mathcal{C}} = d^*_{\mathcal{C}} \geq \sum_{l=0}^{L-1} \left( \overline{b}^{(l)\top} \left( \lambda^{(l)} \right)_+ - \underline{b}^{(l)\top} \left( \lambda^{(l)} \right)_- - b^{(l)\top} \mu^{(l)} \right) + c_0 - \sup_{x \in \mathcal{S}_{in}(x^{\text{nom}})} \left( \mathbf{W}^{(0)\top} \mu^{(0)} \right)^\top x,$$

where the dual variables $(\mu^{[L]}, \lambda^{[L]})$ are determined by a backward propagation

$$\lambda^{(L-1)} = -c, \quad \mu^{(l)} = \overline{a}^{(l)} \left( \lambda^{(l)} \right)_+ + \underline{a}^{(l)} \left( \lambda^{(l)} \right)_-, \quad \lambda^{(l-1)} = \mathbf{W}^{(l)\top} \mu^{(l)} \quad \forall l \in [L-1],$$

We provide the derivation of this algorithm in Appendix G. It turns out that this algorithm can exactly recover the algorithm proposed in Wong and Kolter [2018], where

$$\underline{\sigma}^{(l)}(z^{(l)}) := \alpha^{(l)} z^{(l)}, \quad \overline{\sigma}^{(l)}(z^{(l)}) := \frac{\overline{z}^{(l)}}{\overline{z}^{(l)} - \underline{z}^{(l)}} (z^{(l)} - \underline{z}^{(l)}),$$

and $0 \leq \alpha^{(l)} \leq 1$ represents the slope of the lower bound. When $\alpha^{(l)} = \frac{\overline{z}^{(l)}}{\overline{z}^{(l)} - \underline{z}^{(l)}}$, the greedy algorithm also recovers Fast-Lin [Weng et al., 2018], which explains the arrow from Wong and Kolter [2018] to Weng et al. [2018] in Fig. 1. When $\alpha^{(l)}$ is chosen adaptively as in CROWN [Zhang et al., 2018], the greedy algorithm then recovers CROWN, which explains the arrow from Wong and Kolter [2018] to Zhang et al. [2018] in Fig. 1. See Appendix D for more discussions on the relationship between the primal and dual greedy solvers.

## 5 Optimal LP-relaxed Verification

In the previous sections, we presented a framework that subsumes all existing layer-wise convex-relaxed verification algorithms except that of Raghunathan et al. [2018b]. For ReLU networks, being piece-wise linear, these correspond exactly to the set of all existing LP-relaxed algorithms, as discussed above. We showed the existence of a barrier, $p^*_{\mathcal{C}}$, that limits all such algorithms. Is this just theoretical babbling or is this barrier actually problematic in practice?

In the next section, we perform extensive experiments on deep ReLU networks, evaluating the tightest convex relaxation afforded by our framework (denoted **LP-ALL**) against a greedy dual algorithm (Algorithm 1 of Wong and Kolter [2018], denoted **LP-GREEDY**) as well as another algorithm **LP-LAST**, intermediate in speed and accuracy between them. Both LP-GREEDY and LP-LAST solve the bounds $\underline{z}^{[L]}, \overline{z}^{[L]}$ by setting the dual variables heuristically (see previous section), but LP-GREEDY solves the adversarial loss in the same manner while LP-LAST solves this final LP exactly. We also compare them with the opposite bounds provided by PGD attack [Madry et al., 2017], as well as exact results from MILP [Tjeng et al., 2019] [5].

For the rest of the main text, we are only concerned with ReLU networks, so ($\mathcal{C}$) subject to (4) is in fact an LP.

## 5.1 LP-ALL Implementation Details

In order to exactly solve the tightest LP-relaxed verification problem of a ReLU network, two steps are required: (A) obtaining the tightest pre-activation upper and lower bounds of all the neurons in the NN, excluding those in the last layer, then (B) solving the LP-relaxed verification problem exactly for the last layer of the NN.

**Step A: Obtaining Pre-activation Bounds.** This can be done by solving sub-problems of the orginial relaxed problem ($\mathcal{C}$) subject to (4). Given a NN with $L_0$ layers, for each layer $l_0 \in [L_0]$, we obtain a lower (resp. upper) bound $\underline{z}_j^{(l_0)}$ (resp. $\overline{z}_j^{(l_0)}$) of $z_j^{(l_0)}$, for all neurons $j \in [n^{(l_0)}]$. We do this by setting

$$L \leftarrow l_0, \quad c^\top \leftarrow \mathbf{W}_{j,:}^{(l_0)} \text{ (resp. } c^\top \leftarrow -\mathbf{W}_{j,:}^{(l_0)}\text{)}, \quad c_0 \leftarrow b_j^{(l_0)} \text{ (resp. } c_0 \leftarrow -b_j^{(l_0)}\text{)}$$

in ($\mathcal{C}$) and computing the exact optimum. However, we need to solve an LP for each neuron, and practical networks can have millions of them. We utilize the fact that in each layer $l_0$, computing the bounds $\overline{z}_j^{(l_0)}$ and $\underline{z}_j^{(l_0)}$ for each $j \in [n^{(l_0)}]$ can proceed independently in parallel. Indeed, we design a scheduler to do so on a cluster with 1000 CPU-nodes. See Appendix J for details.

**Step B: Solving the LP-relaxed Problem for the Last Layer.** After obtaining the pre-activation bounds on all neurons in the network using step (A), we solve the LP in ($\mathcal{C}$) subject to (4) for all $j \in [n^{(L_0)}]\backslash\{j^{\text{nom}}\}$ obtained by setting

$$L \leftarrow L_0, \quad c^\top \leftarrow \mathbf{W}_{j^{\text{nom}},:}^{(L_0)} - \mathbf{W}_{j,:}^{(L_0)}, \quad c_0 \leftarrow b_{j^{\text{nom}}}^{(L_0)} - b_j^{(L_0)}$$

again in ($\mathcal{C}$) and computing the exact minimum. Here, $j^{\text{nom}}$ is the true label of the data point $x^{\text{nom}}$ at which we are verifying the network. *We can certify the network is robust around $x^{nom}$ iff the solutions of all such LPs are positive, i.e. we cannot make the true class logit lower than any other logits.* Again, note that these LPs are also independent of each other, so we can solve them in parallel.

Given any $x^{\text{nom}}$, LP-ALL follows steps (A) then (B) to produce a certificate whether the network is robust around a given datapoint or not. LP-LAST on the other hand solves only step (B), and instead of doing (A), it finds the preactivation bounds greedily as in Algorithm 1 of Wong and Kolter [2018].

## 6 Experiments

We conduct two experiments to assess the tightness of LP-ALL: 1) finding certified upper bounds on the robust error of several NN classifiers, 2) finding certified lower bounds on the minimum adversarial distortion $\epsilon$ using different algorithms. All experiments are conducted on MNIST and/or CIFAR-10 datasets.

**Architectures.** We conduct experiments on a range of ReLU-activated feedforward networks. MLP-A and MLP-B refer to multilayer perceptrons: MLP-A has 1 hidden layer with 500 neurons, and MLP-B has 2 hidden layers with 100 neurons each. CNN-SMALL, CNN-WIDE-K, and CNN-DEEP-K are the ConvNet architectures used in Wong et al. [2018]. Full details are in Appendix I.1.

**Training Modes.** We conduct experiments on networks trained with a regular cross-entropy (CE) loss function and networks trained to be robust. These networks are identified by a prefix corresponding to the method used to train them: **LPD** when the LP-relaxed dual formulation of Wong and Kolter [2018] is used for robust training, **ADV** when adversarial examples generated using PGD are used for robust training, as in Madry et al. [2017], and **NOR** when the network is normally trained using the CE loss function. Training details are in Appendix I.2.

**Experimental Setup.** We run experiments on a cluster with 1000 CPU-nodes. The total run time amounts to more than 22 CPU-years. Appendix J provides additional details about the computational resources and the scheduling scheme used, and Appendix K provides statistics of the verification time in these experiments.

Table 1: Certified bounds on the robust error on the test set of MNIST for normally and robustly trained networks. The prefix of each network corresponds to the training method used: ADV for PGD training [Madry et al., 2017], NOR for normal CE loss training, and LPD when the LP-relaxed dual formulation of Wong and Kolter [2018] is used for robust training.

| NETWORK | $\epsilon$ | TEST ERROR | LOWER BOUND | | UPPER BOUND | | |
|---|---|---|---|---|---|---|---|
| | | | PGD | MILP | MILP | LP-ALL | LP-GREEDY |
| ADV-MLP-B | 0.03 | 1.53% | 4.17% | 4.18% | 5.78% | 10.04% | 13.40% |
| ADV-MLP-B | 0.05 | 1.62% | 6.06% | 6.11% | 11.38% | 23.29% | 33.09% |
| ADV-MLP-B | 0.1 | 3.33% | 15.86% | 16.25% | 34.37% | 61.59% | 71.34% |
| ADV-MLP-A | 0.1 | 4.18% | 11.51% | 14.36% | 30.81% | 60.14% | 67.50% |
| NOR-MLP-B | 0.02 | 2.05% | 10.06% | 10.16% | 13.48% | 26.41% | 35.11% |
| NOR-MLP-B | 0.03 | 2.05% | 20.37% | 20.43% | 48.67% | 65.70% | 75.85% |
| NOR-MLP-B | 0.05 | 2.05% | 53.37% | 53.37% | 94.04% | 97.95% | 99.39% |
| LPD-MLP-B | 0.1 | 4.09% | 13.39% | 14.45% | 14.45% | 17.24% | 18.32% |
| LPD-MLP-B | 0.2 | 15.72% | 33.85% | 36.33% | 36.33% | 37.50% | 41.67% |
| LPD-MLP-B | 0.3 | 39.22% | 57.29% | 59.85% | 59.85% | 60.17% | 66.85% |
| LPD-MLP-B | 0.4 | 67.97% | 81.85% | 83.17% | 83.17% | 83.62% | 87.89% |

## 6.1 Certified Bounds on the Robust Error

Table 1 presents the clean test errors and (upper and lower) bounds on the true robust errors for a range of classifiers trained with different procedures on MNIST. For both ADV- and LPD-trained networks, the $\epsilon$ in Table 1 denotes the $l_\infty$-norm bound used for training *and* robust testing; for NORmally-trained networks, $\epsilon$ is only used for the latter.

Lower bounds on the robust error are calculated by finding adversarial examples for inputs that are not robust. This is done by using PGD, a strong first-order attack, or using MILP [Tjeng et al., 2019]. Upper bounds on the robust error are calculated by providing certificates of robustness for input that is robust. This is done using MILP, the dual formulation (LP-GREEDY) presented by Wong and Kolter [2018], or our LP-ALL algorithm.

For the MILP results, we use the code accompanying the paper by Tjeng et al. [2019]. We run the code in parallel on a cluster with 1000 CPU-nodes, and set the MILP solver's time limit to 3600 seconds. Note that this time limit is reached for ADV and NOR, and therefore the upper and lower bounds are separated by a gap that is especially large for some of the NORmally trained networks. On the other hand, for LPD-trained networks, the MILP solver finishes within the time limit, and thus the upper and lower bounds match.

**Results.** For all NORmally and ADV-trained networks, we see that the certified upper bounds using LP-GREEDY and LP-ALL are very loose when we compare the gap between them to the lower bounds found by PGD and MILP. As a sanity check, note that LP-ALL gives a tighter bound than LP-GREEDY in each case, as one would expect. Yet this improvement is not significant enough to close the gap with the lower bounds.

This sanity check also passes for LPD-trained networks, where the LP-GREEDY-certified robust error upper bound is, as expected, much closer to the true error (given by MILP here) than for other networks. For $\epsilon = 0.1$, the improvement of LP-ALL-certified upper bound over LP-GREEDY is at most modest, and the PGD lower bound is tighter to the true error. For large $\epsilon$, the improvement is much more significant in relative terms, but the absolute improvement is only $4 - 7\%$. In this large $\epsilon$ regime, however, both the clean and robust errors are quite large, so the tightness of LP-ALL is less useful.

## 6.2 Certified Bounds on the Minimum Adversarial Distortion $\epsilon$

We are interested in searching for the minimum adversarial distortion $\epsilon$, which is the radius of the largest $l_\infty$ ball in which no adversarial examples can be crafted. An upper bound on $\epsilon$ is calculated using PGD, and lower bounds are calculated using LP-GREEDY, LP-LAST, or our LP-ALL, all via binary search. Since solving LP-ALL is expensive, we find the $\epsilon$-bounds only for ten samples of the MNIST and CIFAR-10 datasets. In this experiment, both ADV- and LPD-networks are trained with an $l_\infty$ maximum allowed perturbation of 0.1 and $8/255$ on MNIST and CIFAR-10, respectively. See Appendix L.1 for details. Fig. 3 and 8 in the Appendix show the median percentage gap (defined in

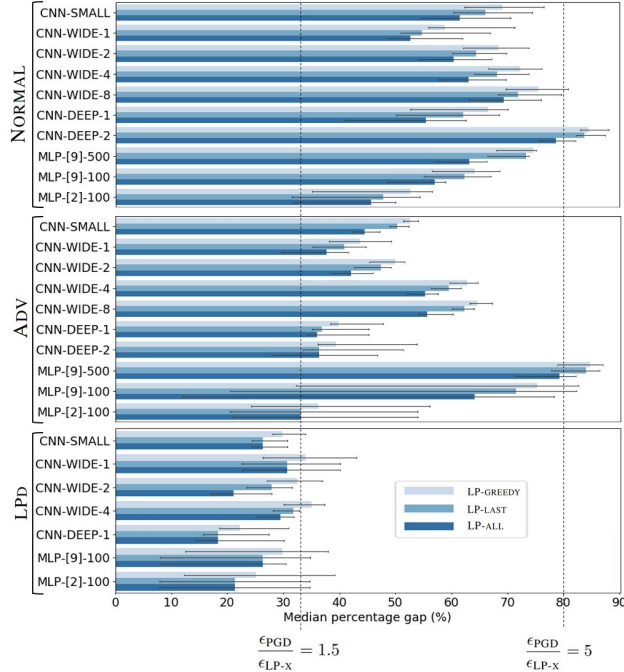

Figure 3: The median percentage gap between the convex-relaxed algorithms (LP-ALL, LP-LAST, and LP-GREEDY) and PGD estimates of the minimum adversarial distortion $\epsilon$ on ten samples of MNIST. The error bars correspond to 95% confidence intervals. We highlight the $1.5\times$ and $5\times$ gaps between the $\epsilon$ value estimated by PGD, and those estimated by the LP-relaxed algorithms. For more details, please refer to Table 2 in Appendix L.2.

Appendix L.2) between the convex-relaxed algorithms and PGD bounds of $\epsilon$ for MNIST and CIFAR, respectively. Details are reported in Tables 2 and 3 in Appendix L.2.

On MNIST, the results show that for all networks trained NORmally or via ADV, the certified lower bounds on $\epsilon$ are 1.5 to 5 times smaller than the upper bound found by PGD; for LPD trained networks, below 1.5 times smaller. On CIFAR-10, the bounds are between 1.5 and 2 times smaller across all models. The smaller gap for LPD is of course as expected following similar observations in prior work [Wong and Kolter, 2018, Tjeng et al., 2019]. Furthermore, the improvement of LP-ALL and LP-LAST over LP-GREEDY is not significant enough to close the gap with the PGD upper bound. Note that similar results hold as well for randomly initialized networks (no training). To avoid clutter, we report these in Appendix M.

## 7 Conclusions and Discussions

In this work, we first presented a layer-wise convex relaxation framework that unifies all previous LP-relaxed verifiers, in both primal and dual spaces. Then we performed extensive experiments to show that even the optimal convex relaxation for ReLU networks in this framework cannot obtain tight bounds on the robust error in all cases we consider here. Thus any method will face a *convex relaxation barrier* as soon as it can be described by our framework. We look at how to bypass this barrier in Appendix A.

Note that different applications have different requirements for the tightness of the verification, so our barrier could be a problem for some but not for others. In so far as the ultimate goal of robustness verification is to construct a training method to lower certified error, this barrier is not necessarily problematic — some such method could still produce networks for which convex relaxation as described by our framework produces accurate robust error bounds. An example is the recent work of Gowal et al. [2018] which shows that interval bound propagation, which often leads to loose certification bounds, can still be used for verified training, and is able to achieve state-of-the-art verified accuracy when carefully tuned. However, without a doubt, in all cases, tighter estimates should lead to better results, and we reveal a definitive ceiling on most current methods.

## Footnotes

[3]The radius of the largest $l_\infty$ ball in which no adversarial examples can be found.

[4]Theorem 2 in Dvijotham et al. [2018b] is a special case of our Theorem 4.2, when applied to ReLU networks. Our proof makes use of the Fenchel-Moreau theorem to deal with general nonlinearities, which is different from that in Dvijotham et al. [2018b].

[5]Note that in practice (as in [Tjeng et al., 2019]), MILP has a time budget, and usually not every sample can be verified within that budget, so that in the end we still obtain only lower and upper bounds given by samples verified to be robust or nonrobust

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
