[Supplementary Material]

# A  How to bypass the barrier?

The primal problem in our framework ($\mathcal{C}$) has several possible sources of looseness:

(1) We relax the nonlinearity $\sigma^{(l)}$ on a box domain $\{\underline{z}^{(l)} \leq z \leq \overline{z}^{(l)}\}$. This relaxation is simple to perform, but might come at the cost of losing some correlations between the coordinates of $z$ and of obtaining a looser relaxation. Note that our framework does consider the correlations between coordinates of $z^{(l)}$ to get bounds for all later layers, however it relies on $\underline{z}^{(l)}$ and $\overline{z}^{(l)}$ which are considered individually, without interactions within the same layer.

(2) We solve for the bounds $\underline{z}^{[l]}, \overline{z}^{[l]}$ recursively, and we incur some gap for every recursion; a loose bound in earlier layers will make bounds for later layers even looser. This can be problematic for very deep networks or recurrent networks.

(3) In the specific case of ReLU, we lose a bit every time we relax over an unstable neuron; one possible future direction is to combine branch-and-bound with convex relaxation to strategically split the domains of unstable neurons.

Any method that improves on any of the above issues can possibly bypass the barrier; see, e.g., SDP-based verifiers [Raghunathan et al., 2018b] can consider the interaction between each neuron within one layer; [Anderson et al., 2018] can relax the combination of one ReLU layer and one affine layer. On the other hand, exact verifiers [Katz et al., 2017, Ehlers, 2017], local Lipschitz-constant-based verifiers [Zhang et al., 2019, Raghunathan et al., 2018a], and hybrid approaches [Bunel et al., 2018, Singh et al., 2019b] do not fall under the purview of our framework. In general, none of them are strictly better than the convex relaxation approach and they make trading-offs between speed and accuracy. However, it would be fruitful to consider combinations of these methods in the future, as done in Singh et al. [2019b]. We hope our work will foster much thought in the community toward new relaxation paradigms for tight neural network verification.

# B  The optimal layer-wise convex relaxation

## B.1  The optimal convex relaxation of a single nonlinear neuron

In this section, we give the optimal convex relaxation of a single nonlinear neuron $x = \sigma(z)$, which is the convex hull of its graph. Although the proof is elementary, we provide it for completeness.

**Proposition B.1.** *Suppose the activation function $\sigma : [\underline{z}, \overline{z}] \subset \mathbb{R}^{n_z} \to \mathbb{R}$ is bounded from above and below. Let $\underline{\sigma}_{opt}$ and $-\overline{\sigma}_{opt}$ be the greatest closed convex functions majored by $\sigma$ and $-\sigma$, respectively, i.e.,*

$$\underline{\sigma}_{opt}(z) := \sup_{(\alpha,\gamma)\in\mathcal{A}} \alpha^\top z + \gamma, \quad \text{where } \mathcal{A} = \{(\alpha,\gamma) : \alpha^\top z' + \gamma \leq \sigma(z'), \forall z' \in [\underline{z}, \overline{z}]\},$$

$$\overline{\sigma}_{opt}(z) := \inf_{(\alpha,\gamma)\in\mathcal{A}'} \alpha^\top z + \gamma, \quad \text{where } \mathcal{A}' = \{(\alpha,\gamma) : \alpha^\top z' + \gamma \geq \sigma(z'), \forall z' \in [\underline{z}, \overline{z}]\} \tag{12}$$

*Then we have,*

*1. Both $\underline{\sigma}_{opt}$ and $\overline{\sigma}_{opt}$ are continuous in $[\underline{z}, \overline{z}]$.*

*2.*

$$\big\{(z,x) : \underline{\sigma}_{opt}(z) \leq x \leq \overline{\sigma}_{opt}(z), \underline{z} \leq z \leq \overline{z}\big\} = \overline{conv}\big(\{(z,x) : x = \sigma(z), \underline{z} \leq z \leq \overline{z}\}\big),$$

*where $\overline{conv}$ denotes the closed convex hull.*

*Proof.*

1. By the boundedness of $\sigma$ on $[\underline{z}, \overline{z}]$, we know that the effective domain of $\underline{\sigma}_{\text{opt}}$ and $-\overline{\sigma}_{\text{opt}}$ is $[\underline{z}, \overline{z}]$. By definition (12), $\underline{\sigma}_{\text{opt}}$ and $-\overline{\sigma}_{\text{opt}}$ are closed convex functions. By Theorem 10.2 in Rockafellar [2015], we know that both $\underline{\sigma}_{\text{opt}}$ and $-\overline{\sigma}_{\text{opt}}$ are continuous in $[\underline{z}, \overline{z}]$, so is $\overline{\sigma}_{\text{opt}}$.

2. We first decompose the left-hand-side into 3 terms:

$$\big\{(z,x) : \underline{\sigma}_{\text{opt}}(z) \leq x \leq \overline{\sigma}_{\text{opt}}(z), \underline{z} \leq z \leq \overline{z}\big\} = \big\{\underline{\sigma}_{\text{opt}}(z) \leq x\big\} \cap \big\{x \leq \overline{\sigma}_{\text{opt}}(z)\big\} \cap \big\{\underline{z} \leq z \leq \overline{z}\big\}.$$

Let $\underline{\mathcal{F}} = \{(\alpha, \gamma) : \alpha^T z' + \gamma \leq \sigma(z'), \forall z' \in [\underline{z}, \overline{z}]\}$ and $\overline{\mathcal{F}} = \{(\alpha, \gamma) : \alpha^T z' + \gamma \geq \sigma(z'), \forall z' \in [\underline{z}, \overline{z}]\}$. For the first term, by definition (12) we have

$$\{\underline{\sigma}_{\text{opt}}(z) \leq x\} = \cap_{\{(\alpha, \gamma) : \alpha^T z' + \gamma \leq \sigma(z'), \forall z' \in [\underline{z}, \overline{z}]\}} \{\alpha^T z + \gamma \leq x\}$$
$$= \cap_{\{(\alpha, \beta, \gamma) : \beta < 0, \alpha^T z' + \beta \sigma(z') + \gamma \leq 0, \forall z' \in [\underline{z}, \overline{z}]\}} \{\alpha^T z + \beta x + \gamma \leq 0\}.$$

For the second term, by definition (12) we have

$$\{x \leq \overline{\sigma}_{\text{opt}}(z)\} = \cap_{\{(\alpha, \gamma) : \alpha^T z' + \gamma \leq -\sigma(z'), \forall z' \in [\underline{z}, \overline{z}]\}} \{\alpha^T z + \gamma \leq -x\}$$
$$= \cap_{\{(\alpha, \beta, \gamma) : \beta > 0, \alpha^T z' + \beta \sigma(z') + \gamma \leq 0, \forall z' \in [\underline{z}, \overline{z}]\}} \{\alpha^T z + \beta x + \gamma \leq 0\}.$$

For the third term, we have

$$\{\underline{z} \leq z \leq \overline{z}\} = \cap_{\{(\alpha, \gamma) : \alpha^T z' + \gamma \leq 0, \forall z' \in [\underline{z}, \overline{z}]\}} \{\alpha^T z + \gamma \leq 0\}$$
$$= \cap_{\{(\alpha, \beta, \gamma) : \beta = 0, \alpha^T z' + \beta \sigma(z') + \gamma \leq 0, \forall z' \in [\underline{z}, \overline{z}]\}} \{\alpha^T z + \beta x + \gamma \leq 0\}.$$

Combining the three terms, we conclude the proof by

$$\{(z, x) : \underline{\sigma}_{\text{opt}}(z) \leq x \leq \overline{\sigma}_{\text{opt}}(z), \underline{z} \leq z \leq \overline{z}\}$$
$$= \cap_{\{(\alpha, \beta, \gamma) : \alpha^T z' + \beta \sigma(z') + \gamma \leq 0, \forall z' \in [\underline{z}, \overline{z}]\}} \{\alpha^T z + \beta x + \gamma \leq 0\}$$
$$= \overline{\text{conv}}\big(\{(z, x) : x = \sigma(z), \underline{z} \leq z \leq \overline{z}\}\big),$$

where we use the definition of closed convex hull in the last identity.

$\square$

## B.2 The optimal convex relation of a nonlinear layer

When $x^{(l+1)} = \sigma^{(l)}(z^{(l)})$ is a nonlinear layer that has a vector output $x^{(l+1)} \in \mathbb{R}^{n^{(l+1)}}$, the optimal convex relaxation may not have a simple analytic form as $\underline{\sigma}_{\text{opt}}^{(l)}(z^{(l)}) \leq x^{(l+1)} \leq \overline{\sigma}_{\text{opt}}^{(l)}(z^{(l)})$. Fortunately, if there is no interaction (as defined below) among the output neurons, the optimal convex relaxation can be given as a simple analytic form.

**Definition B.2** (non-interactive layer). *Let $\sigma : \mathbb{R}^m \to \mathbb{R}^n$ and $x = \sigma(z)$ be a nonlinear layer with input $z \in [\underline{z}, \overline{z}] \subset \mathbb{R}^m$ and output $x \in \mathbb{R}^n$. For each output $x_j$, let $I_j \subset [m]$ be the minimal set of z's entries that affect $x_j$, where $x_j = \sigma(z_{I_j})$. We call the layer $x = \sigma(z)$ non-interactive if the sets $I_j$ ($j \in [n]$) are mutually disjoint.*

Commonly used nonlinear activation layers are all non-interactive. It is obvious that all entry-wise nonlinear layers, such as (leaky-)ReLU and sigmoid, are non-interactive. A MaxPool layer with non-overlapping regions (stride no smaller than kernel size) is also non-interactive. Finally, any layer with scalar-valued output is non-interactive. When we treat a general nonlinear specification (as proposed in Qin et al. [2019]) as an additional nonlinear layer $x^{(L+1)} = F(x^{(0)}, x^{(L)})$, this layer is automatically non-interactive. This nice property ensures that our framework can deal with very general specifications.

The optimal convex relaxation of a non-interactive layer has a simple analytic form as below.

**Proposition B.3.** *If the layer $\sigma^{(l)} : [\underline{z}^{(l)}, \overline{z}^{(l)}] \to \mathbb{R}^{n^{(l+1)}}$ is non-interactive, we have*

$$\{(z^{(l)}, x^{(l+1)}) : \underline{\sigma}_{opt}^{(l)}(z^{(l)}) \leq x^{(l+1)} \leq \overline{\sigma}_{opt}^{(l)}(z^{(l)})\} =$$
$$\overline{conv}\big(\{(z^{(l)}, x^{(l+1)}) : x^{(l+1)} = \sigma^{(l)}(z^{(l)}), \quad \underline{z}^{(l)} \leq z^{(l)} \leq \overline{z}^{(l)}\}\big),$$

*where $\overline{conv}$ denotes the closed convex hull, and vector-valued functions $\underline{\sigma}_{opt}^{(l)}(z)$ and $\overline{\sigma}_{opt}^{(l)}(z)$ are defined in (6) for each output entry.*

Thanks to its non-interaction, Proposition B.3 is a direct consequence of item 2 in Proposition B.1.

## C Convex relaxations not included in Problem ($\mathcal{C}$).

We emphasize that by *optimal*, we mean the optimal convex relaxation of the *single* nonlinear constraint $x^{(l+1)} = \sigma^{(l)}(z^{(l)})$ (see Proposition (B.3)) instead of the optimal convex relaxation of the

nonconvex feasible set of the original problem ($\mathcal{O}$). In fact, for neural networks with more than two hidden layers ($L \geq 2$), the optimal convex relaxation of the nonconvex feasible set of problem ($\mathcal{O}$) is a strict subset of the feasible set of problem ($\mathcal{C}$), even with the tightest bounds ($\underline{z}^{[L]}, \overline{z}^{[L]}$) and the optimal choice of $\underline{\sigma}_{\text{opt}}^{(l)}(z)$ and $\overline{\sigma}_{\text{opt}}^{(l)}(z)$ in (6). It is possible to obtain other (maybe tighter) convex relaxations [Anderson et al., 2018], but it comes with more assumptions on the nonlinear layers and more complex convex constraints.

For example, Raghunathan et al. [2018b] rewrites the ReLU nonlinearity as a quadratic constraint, and then proposes a semidefinite programming (SDP) relaxation for the resulting quadratic optimization problem. Problem ($\mathcal{C}$) does not cover this SDP-relaxation. Sometimes Problem ($\mathcal{C}$) provides tighter relaxation than the SDP-relaxation, e.g., the case when there is only one neuron in a layer, while sometimes the SDP-relaxation provides tighter relaxation than Problem ($\mathcal{C}$), e.g., the examples provided in Raghunathan et al. [2018b]. The SDP-relaxation currently only works for ReLU nonlinearity. It is not clear to us how to extend the SDP-relaxed verifier to general nonlinearities. On the other hand, Problem ($\mathcal{C}$) can handle any non-interactive nonlinear layer and any nonlinear specification.

## D  Greedily solving the primal with linear bounds.

In this section, we show how to greedily solve ($\mathcal{C}$) by over-relaxing the problem to give a lower bound directly, and discuss the relationships between algorithms in Figure 1, especially for the algorithms in primal view.

**Relaxing the ReLU neurons.**   We start with giving exactly one linear upper bound and exactly one linear lower bound for each activation function in ($\mathcal{C}$):

$$
\begin{aligned}
\min_{(x^{[L+1]}, z^{[L]}) \in \mathcal{D}} \quad & c^\top x^{(L)} + c_0 \\
\text{s.t.} \quad & z^{(l)} = \mathbf{W}^{(l)} x^{(l)} + b^{(l)}, \quad l \in \{0, \cdots, L-1\}, \\
\underline{a}^{(l)} z^{(l)} + \underline{b}^{(l)} \leq & x^{(l+1)} \leq \overline{a}^{(l)} z^{(l)} + \overline{b}^{(l)}, \quad l \in \{0, \cdots, L-1\},
\end{aligned}
\tag{13}
$$

Typically, the selection of $\underline{a}^{(l)}$, $\overline{a}^{(l)}$, $\overline{b}^{(l)}$, $\underline{b}^{(l)}$ can depend on $\overline{z}^{(l)}$ and $\underline{z}^{(l)}$ to minimize the error between the upper/lower bound and the activation function. For element-wise activation functions, the linear upper and lower bounds are usually also element-wise. For example, for an unstable ReLU neuron with $\overline{z}_i^{(l)} > 0$ and $\underline{z}_i^{(l)} < 0$, one upper bound is $x_i^{(l+1)} \leq \frac{\overline{z}_i^{(l)}}{\overline{z}_i^{(l)} - \underline{z}_i^{(l)}} z_i^{(l)} - \frac{\overline{z}_i^{(l)} \underline{z}_i^{(l)}}{\overline{z}_i^{(l)} - \underline{z}_i^{(l)}}$. According to Proposition B.1, this is the optimal convex relaxation for the upper bound. For the lower bound, the optimal convex relaxation $(x_i^{(l+1)} \geq z_i^{(l)}) \cap (x_i^{(l+1)} \geq 0)$ is not achievable as one linear function; we use any over-relaxed bounds $x_i^{(l+1)} \geq \underline{a}_i^{(l)} z_i^{(l)}$ with $0 \leq \underline{a}_i^{(l)} \leq 1$ as the lower bound. This perspective covers Fast-Lin [Weng et al., 2018], DeepZ [Singh et al., 2018] and Neurify [Wang et al., 2018b], where the lower bound is fixed as $\underline{a}_i^{(l)} = \overline{a}_i^{(l)} = \frac{\overline{z}_i^{(l)}}{\overline{z}_i^{(l)} - \underline{z}_i^{(l)}}$; this is referred as a "zonotope" relaxation in AI$^2$ [Gehr et al., 2018] and DeepZ. AI$^2$ is a general technique of using "abstract transformers" (sound relaxations of neural network elements) to verify neural networks, but it uses suboptimal relaxations for ReLU non-linearity; DeepZ further refines the transformers for ReLU and significantly outperforms AI$^2$ [Singh et al., 2018]. Other activation functions can be linearly bounded as discussed in CROWN [Zhang et al., 2018], DeepZ and DeepPoly [Singh et al., 2019a]; CROWN and DeepPoly are also more general and do not require $\underline{a}_i^{(l)} = \overline{a}_i^{(l)}$ to allow a more flexible selection of bounds.

**Deriving the Greedy Primal Method.**   Assuming we have obtained the linear upper and lower bounds for $x^{(l+1)}$ with respect to $z^{(l)}$, $\underline{z}_i^{(l+1)}$ can be formed greedily as a linear combination of these linear bounds: we greedily select the upper bound $x_i^{(l+1)} \leq \underline{a}_i^{(l)} z_i^{(l)} + \underline{b}_i^{(l)}$ when $\mathbf{W}_{i,k}^{(l+1)}$ is negative, and select the lower bound $x_i^{(l+1)} \geq \overline{a}_i^{(l)} z_i^{(l)} + \overline{b}_i^{(l)}$ otherwise. This bound reflects the worst case scenario without considering any other neurons:

$$
z_i^{(l+1)} \geq \underline{z}_i^{(l+1)} := \underline{A}_{i,:}^{(l)} z^{(l)} + \underline{b}_i'^{(l)}
\tag{14}
$$

where matrix $\underline{A}_{i,k}^{(l)} = \begin{cases} \mathbf{W}_{i,k}^{(l+1)}\overline{a}_k^{(l)}, & \mathbf{W}_{i,k}^{(l+1)} < 0 \\ \mathbf{W}_{i,k}^{(l+1)}\underline{a}_k^{(l)}, & \mathbf{W}_{i,k}^{(l+1)} \geq 0 \end{cases}$ reflects the chosen upper or lower bound based

on the sign of $\mathbf{W}_{i,k}^{(l+1)}$, and vector $\underline{b}_i^{\prime(l)} = \sum_{k,\mathbf{W}_{i,k}^{(l+1)}\geq 0} \mathbf{W}_{i,k}^{(l+1)}\underline{b}_k^{(l)} + \sum_{k,\mathbf{W}_{i,k}^{(l+1)}<0} \mathbf{W}_{i,k}^{(l+1)}\overline{b}_k^{(l)} + b_i^{(l)}$.

The lower bound $\underline{z}_i^{(l+1)}$ can also be formed similarly. Eventually, we get one linear upper bound and one linear lower bound for $z_i^{(l+1)}$, written as:

$$\underline{A}_{i,:}^{(l)}z^{(l)} + \underline{b}_i^{\prime(l)} \leq z_i^{(l+1)} \leq \overline{A}_{i,:}^{(l)}z^{(l)} + \overline{b}_i^{\prime(l)} \tag{15}$$

A sharp-eyed reader can notice that it is possible to also get a similar bound for each component of $z^{(l)}$ and plug it into (15), thus obtaining a linear upper bound and a linear lower bound for $z_i^{(l+1)}$ with respect to $z^{(l-1)}$. To do this, we first substitute $z^{(l)} = \mathbf{W}^{(l)}x^{(l)} + b^{(l)}$ into Eq. (15), obtaining

$$\underline{A}_{i,:}^{(l)}(\mathbf{W}^{(l)}x^{(l)} + b^{(l)}) + \underline{b}_i^{\prime(l)} \leq z_i^{(l+1)} \leq \overline{A}_{i,:}^{(l)}(\mathbf{W}^{(l)}x^{(l)} + b^{(l)}) + \overline{b}_i^{\prime(l)}$$

Applying the bounds on $x^{(l)}$ with respect to $z^{(l-1)}$, and using a similar technique as we did above to obtain (15), we get linear upper and lower bounds for $z_i^{(l+1)}$ with respect to $z^{(l-1)}$ in the following form:

$$\underline{A}_{i,:}^{(l-1)}z^{(l-1)} + \underline{b}_i^{\prime(l-1)} \leq z_i^{(l+1)} \leq \overline{A}_{i,:}^{(l-1)}z^{(l-1)} + \overline{b}_i^{\prime(l-1)} \tag{16}$$

where $\underline{b}_i^{\prime(l-1)}$ and $\overline{b}_i^{\prime(l-1)}$ collect all bias terms in the substitution process. Caution has to be taken when forming $\underline{A}_{i,:}^{(l-1)}$ and $\overline{A}_{i,:}^{(l-1)}$, as we need to choose $\underline{a}_k^{(l-1)}$ or $\overline{a}_k^{(l-1)}$ based on the sign of $\underline{A}_{i,:}^{(l)}\mathbf{W}_{:,k}^{(l)}$, since the coefficients before each inequality now become $\underline{A}_{i,:}^{(l)}\mathbf{W}^{(l)}$ rather than just $\mathbf{W}^{(l)}$:

$$\underline{A}_{i,k}^{(l-1)} = \begin{cases} \underline{A}_{i,:}^{(l)}\mathbf{W}_{:,k}^{(l)}\overline{a}_k^{(l-1)}, & \underline{A}_{i,:}^{(l)}\mathbf{W}_{:,k}^{(l)} < 0 \\ \underline{A}_{i,:}^{(l)}\mathbf{W}_{:,k}^{(l)}\underline{a}_k^{(l-1)}, & \underline{A}_{i,:}^{(l)}\mathbf{W}_{:,k}^{(l)} \geq 0 \end{cases}$$

An eagle-eyed reader can notice that we can continue this process until we have reached $z^{(0)}$, and obtain the following linear bounds:

$$\underline{A}_{i,:}^{(0)}z^{(0)} + \underline{b}_i^{\prime(0)} \leq z_i^{(l+1)} \leq \overline{A}_{i,:}^{(0)}z^{(0)} + \overline{b}_i^{\prime(0)} \tag{17}$$

where $\underline{A}_{i,:}^{(0)}, \overline{A}_{i,:}^{(0)}, \underline{b}_i^{\prime(0)}$ and $\overline{b}_i^{\prime(0)}$ can be formed similarly as above. Substituting $z^{(0)} = \mathbf{W}^{(0)}x^{(0)} + b^{(0)}$ ($x^{(0)} = x$ is the input of the neural network) simply yields:

$$\underline{A}_{i,:}x + \underline{b}_i' \leq z_i^{(l+1)} \leq \overline{A}_{i,:}x + \overline{b}_i' \tag{18}$$

where $\underline{A}_{i,:} = \underline{A}_{i,:}^{(0)}\mathbf{W}^{(0)}, \overline{A}_{i,:} = \overline{A}_{i,:}^{(0)}\mathbf{W}^{(0)}$ captures the products of $\mathbf{W}$ of all layers and the chosen $\underline{a}_k^{(l)}$ or $\overline{a}_k^{(l)}$ for each layer; $\underline{b}', \overline{b}'$ collects all bias terms (we refer the readers to Theorem 3.2 in Zhang et al. [2018] for the exact form of $\underline{A}, \overline{A}, \underline{b}', \overline{b}'$). This procedure beautifully works as the linear combination of linear bounds are still linear bounds. Eq. (18) is a remarkable result, as the output of a non-linear function (neural network) has been directly bounded linearly for all $x$ close to $x^{\text{nom}}$. This allows us to immediately give upper and lower bounds of $z_i^{(l+1)}$ by considering the worst case $x \in \mathcal{S}_{in}(x^{\text{nom}})$. When the set is an $L_\infty$ normed ball, this is obvious,

$$-\epsilon\|\underline{A}_{i,:}\|_1 + \underline{A}_{i,:}x^{\text{nom}} + \underline{b}_i' \leq z_i^{(l+1)} \leq \epsilon\|\overline{A}_{i,:}\|_1 + \overline{A}_{i,:}x^{\text{nom}} + \overline{b}_i', \tag{19}$$

The entire bound propagation process does not involve any LP solver, so it is efficient and can scale to quite large networks. The final objective $c^\top x^{(L)} + c_0$ can be treated as an additional linear layer after $z^{(L-1)}$. Because to form the bounds for $z^{(L-1)}$ we need to compute bounds for all $z^{(l)}, l \in [L-1]$ beforehand, each in $O(l)$ time, the time complexity of this method is quadratic in $L$.

**Connections Between Existing Methods.** For each neuron, the selection of linear bounds are completely independent; this allows further improvements in this greedy algorithm. For example, the selection of $\underline{a}_k^{(l)}$ can depend on $\overline{z}_k^{(l)}$ and $\underline{z}_k^{(l)}$ to adaptively minimize the error between the lower bound and ReLU function. CROWN [Zhang et al., 2018] and DeepPoly [Singh et al., 2019a] used this strategy to achieve tighter verification results than Fast-Lin [Wang et al., 2018b], DeepZ [Singh et al., 2018] and Neurify [Wang et al., 2018b]. Note that although the bound propagation techniques used in these works can be viewed as using different linear relaxations and solve the primal problem greedily in our framework, each of the works has some unique features. For example, DeepPoly [Singh et al., 2019a] and DeepZ [Singh et al., 2018] carefully consider floating-point rounding during the computation; Weng et al. [2018] gives a theoretical hardness proof based on a reduction from the set-cover problem; Neurify [Wang et al., 2018b] combines the relaxed bound with a branch-and-bound search to give concrete instances of adversarial example if they exist, and also uses the bound for training [Wang et al., 2018a].

One the other hand, instead of propagating the bounds of $z^{(l+1)}$ to $z^{(l-k)}$ as shown above, we can decouple layer $z^{(l-(k-1))}$ and $z^{(l-k)}$ entirely: suppose we have obtained concrete upper and lower bounds for $z^{(l-(k-1))}$, we can treat $z^{(l-(k-1))}$ as the input layer and only consider a $k$-layer network to compute the bounds of $z^{(l+1)}$. This leads to interval bounds propagation (IBP) [Gowal et al., 2018] ($k = 1$) and "Box Domain" [Mirman et al., 2018] which gives even looser bounds, but its computation cost is also greatly reduced.

The greedy algorithm in primal space is also closely connected to the greedy algorithm in dual space; the dual of (13) will recover a dual formulation with solution (47), and the closed from solution are related to the chosen slopes $\overline{a}_i^{(l)}$ and $\underline{a}_i^{(l)}$. This explains the equivalence of Fast-Lin and the greedy algorithm to solve the dual problem presented in Algorithm 1 of [Wong and Kolter, 2018].

**The Relationships Between Algorithms in Figure 1.** Based on the above discussions, we now revisit Figure 1, and discuss each arrow in this figure on the "primal view" side.

First of all, the arrow from "Optimal layer-wise convex relaxation" to CROWN [Zhang et al., 2018] trivially holds since CROWN is a greedy algorithm to solve LP relaxations (problem $\mathcal{C}$ plus Eq. (7)), which can be included in the convex relaxation framework. Additionally, CROWN is proposed as a more general variant of Fast-Lin [Weng et al., 2018]. In Fast-Lin, the linear relaxation uses the same slope for the upper and lower bounds; in CROWN, the slopes can be different. In other words, in Eq. (7), $\overline{a}^{(l)} = \underline{a}^{(l)}$ for Fast-Lin but this is not a requirement for CROWN.

Despite originating from different perspectives, DeepZ [Singh et al., 2018] and Fast-Lin [Weng et al., 2018] share the same relaxations and give numerically identical bounds; so do DeepPoly [Singh et al., 2019a] and CROWN. This can be observed by translating between the different notations of these papers. Particularly, Singh et al. [2019b] commented "DeepZ has *the same* precision as Fast-Lin and DeepPoly has *the same* precision as CROWN", although they have several implementation differences.

The arrows from "LP-Relaxed Dual" to CROWN and "LP-Relaxed Dual" to Fast-Lin come from equation (7), where CROWN and Fast-Lin use one linear upper bound and one linear lower bound as constraints instead of the general convex constraints in ($\mathcal{C}$), so the problem $\mathcal{C}$ becomes a special case of an LP-relaxed problem.

Fast-Lin and Neurify [Wang et al., 2018b] use the same relaxation for ReLU neurons (and unlike other works, these two only deal with ReLU activation functions). This can be observed by comparing Figure 3 in Wang et al. [2018b] and Figure 1 in Weng et al. [2018]: the choice of the slopes $\underline{a}^{(l)}$ and $\overline{a}^{(l)}$ are the same. Numerically, both algorithms also produce the same results, but Neurify additionally implements a branch-and-bound search for solving the exact verification problem with the relaxation based bounds.

Figure 4: Illustration of strong duality proof for a convex problem. Left: proof under Slater's condition (picture from [Boyd and Vandenberghe, 2004] Section 5.3.2). Right: our proof. In both settings, the set $\mathcal{A}$ and $\mathcal{B}$ are convex and do not intersect, so they can be separated by a hyperplane. Slater's condition (Left) assumes existence of a point that strictly satisfies the inequality constraints, i.e., $(\widetilde{u}, \widetilde{t})$ in Figure 5(left), and thus any separating hyperplane must be nonvertical. In our setting (Right), we take $B$ to be a much larger set (thanks to Lemma E.1), and thus any separating hyperplane must be nonvertical. Therefore, we can get strong duality without the Slater's condition.

# E   Strong duality for Problem ($\mathcal{C}$): $p_{\mathcal{C}}^* = d_{\mathcal{C}}^*$

Consider the following perturbed version of problem ($\mathcal{C}$):

$$
\begin{aligned}
\widetilde{p}_{\mathcal{C}}^* := \min_{(x^{[L+1]}, z^{[L]}) \in \mathcal{D}} \quad & c^\top x^{(L)} + c_0 \\
\text{s.t.} \quad & z^{(l)} = \mathbf{W}^{(l)} x^{(l)} + b^{(l)} + v^{(l)}, l \in [L], \\
& \underline{\sigma}^{(l)}(z^{(l)}) - \underline{u}^{(l)} \le x^{(l+1)} \le \overline{\sigma}^{(l)}(z^{(l)}) + \overline{u}^{(l)}, l \in [L].
\end{aligned}
\tag{20}
$$

**Lemma E.1.** *We assume that for each $l \in [L]$, both $\underline{\sigma}^{(l)}$ and $\overline{\sigma}^{(l)}$ have a finite Lipschitz constant in the domain $[\underline{z}^{(l)}, \overline{z}^{(l)}]$. There exists a positive constant $C_{\mathcal{C}} > 0$ such that for any perturbations $(\underline{u}^{[L]}, \overline{u}^{[L]}, v^{[L]})$, we have*

$$
\widetilde{p}_{\mathcal{C}}^* \ge p_{\mathcal{C}}^* - C_{\mathcal{C}} \|(\underline{u}^{[L]}, \overline{u}^{[L]}, v^{[L]})\|_2
\tag{21}
$$

Lemma E.1 shows that the optimal value of the perturbed problem, i.e., $\widetilde{p}_{\mathcal{C}}^*$, "smoothly" changes with the perturbations. We delay the proof of Lemma E.1 in Section E.2. Combined with convexity, this ensures the strong duality for problem ($\mathcal{C}$).

*Proof of Theorem 4.1.* The structure of the proof follows the proof of strong duality given the Slater's condition in [Boyd and Vandenberghe, 2004] (Section 5.3.2). However, we do not assume the Slater's condition in our result here. Let's define

$$
\begin{aligned}
\mathcal{A} = \big\{ (\underline{u}^{[L]}, \overline{u}^{[L]}, v^{[L]}, t) : & \exists (x^{[L+1]}, z^{[L]}) \in \mathcal{D}, \underline{\sigma}^{(l)}(z^{(l)}) - \underline{u}^{(l)} \le x^{(l+1)} \le \overline{\sigma}^{(l)}(z^{(l)}) + \overline{u}^{(l)}, \\
& z^{(l)} = \mathbf{W}^{(l)} x^{(l)} + b^{(l)} + v^{(l)}, \forall l \in [L], c^\top x^{(L)} + c_0 \le t \big\},
\end{aligned}
$$

and

$$
\mathcal{B} = \big\{ (\underline{u}^{[L]}, \overline{u}^{[L]}, v^{[L]}, t) : \quad t < p_{\mathcal{C}}^* - C_{\mathcal{C}} \|(\underline{u}^{[L]}, \overline{u}^{[L]}, v^{[L]})\|_2 \big\}.
$$

$\mathcal{A}$ is convex because the problem (20) is convex. $\mathcal{B}$ is convex by definition. The sets $\mathcal{A}$ and $\mathcal{B}$ do not intersect, as illustrated in Figure 4. To see this, suppose $(\underline{u}^{[L]}, \overline{u}^{[L]}, v^{[L]}, t) \in \mathcal{A} \cap \mathcal{B}$. Since $(\underline{u}^{[L]}, \overline{u}^{[L]}, v^{[L]}, t) \in \mathcal{B}$, we have $t < p_{\mathcal{C}}^* - C_{\mathcal{C}} \|(\underline{u}^{[L]}, \overline{u}^{[L]}, v^{[L]})\|_2$. Since $(\underline{u}^{[L]}, \overline{u}^{[L]}, v^{[L]}, t) \in \mathcal{A}$, there exists $(x^{[L+1]}, z^{[L]}) \in \mathcal{D}$ such that it satisfies the constraints in problem (20), and $t \ge c^\top x^{(L)} + c_0 \ge \widetilde{p}_{\mathcal{C}}^* \ge p_{\mathcal{C}}^* - C_{\mathcal{C}} \|(\underline{u}^{[L]}, \overline{u}^{[L]}, v^{[L]})\|_2$, where the last inequality comes from (21). This is a contradiction!

By the separating hyperplane theorem, there exists $(\underline{\lambda}^{[L]}, \overline{\lambda}^{[L]}, \mu^{[L]}, \nu) \neq 0$ and $\alpha$ such that

$$(\underline{u}^{[L]}, \overline{u}^{[L]}, v^{[L]}, t) \in \mathcal{A} \Rightarrow \underline{\lambda}^{[L]\top} \underline{u}^{[L]} + \underline{\lambda}^{[L]\top} \overline{u}^{[L]} + \mu^{[L]\top} v^{[L]} + \nu t \geq \alpha, \tag{22}$$

and

$$(\underline{u}^{[L]}, \overline{u}^{[L]}, v^{[L]}, t) \in \mathcal{B} \Rightarrow \underline{\lambda}^{[L]\top} \underline{u}^{[L]} + \underline{\lambda}^{[L]\top} \overline{u}^{[L]} + \mu^{[L]\top} v^{[L]} + \nu t \leq \alpha, \tag{23}$$

From (22), we conclude that $\underline{\lambda}^{[L]} \geq 0$, $\overline{\lambda}^{[L]} \geq 0$ and $\nu \geq 0$. Otherwise, $\underline{\lambda}^{[L]\top} \underline{u}^{[L]} + \underline{\lambda}^{[L]\top} \overline{u}^{[L]} + \nu t$ is unbounded from below over $\mathcal{A}$, contradicting (22). Since $(0, 0, 0, t) \in \mathcal{B}$ for any $t < p_{\mathcal{C}}^*$, we have $\nu t \leq \alpha$ for any $t < p_{\mathcal{C}}^*$ thanks to (23), and thus $\nu p_{\mathcal{C}}^* \leq \alpha$. Together with (22), we conclude that for any $(x^{[L+1]}, z^{[L]}) \in \mathcal{D}$,

$$\begin{aligned}
\nu(c^\top x^{(L)} + c_0) &+ \sum_{l=0}^{L-1} \mu^{(l)\top}(z^{(l)} - \mathbf{W}^{(l)} x^{(l)} - b^{(l)}) + \sum_{l=0}^{L-1} \underline{\lambda}^{(l)\top}(\underline{\sigma}^{(l)}(z^{(l)}) - x^{(l+1)}) \\
&+ \sum_{l=0}^{L-1} \overline{\lambda}^{(l)\top}(x^{(l+1)} - \overline{\sigma}^{(l)}(z^{(l)})) \geq \alpha \geq \nu p_{\mathcal{C}}^*.
\end{aligned} \tag{24}$$

Assume that $\nu > 0$. In that case, we can divide (24) by $\nu$ to obtain

$$L(x^{[L+1]}, z^{[L]}, \underline{\lambda}^{[L]}/\nu, \overline{\lambda}^{[L]}/\nu, \mu^{[L]}/\nu) \geq p_{\mathcal{C}}^*$$

for all $(x^{[L+1]}, z^{[L]}) \in \mathcal{D}$, where $L(\cdot)$, defined in (8), is the Lagrangian of $(\mathcal{C})$. Minimizing over $(x^{[L+1]}, z^{[L]}) \in \mathcal{D}$, we obtain $g_{\mathcal{C}}(\mu^{[L]}/\nu, \underline{\lambda}^{[L]}/\nu, \overline{\lambda}^{[L]}/\nu) \geq p_{\mathcal{C}}^*$. By weak duality, we have $g_{\mathcal{C}}(\mu^{[L]}/\nu, \underline{\lambda}^{[L]}/\nu, \overline{\lambda}^{[L]}/\nu) \leq p_{\mathcal{C}}^*$, so in fact $g_{\mathcal{C}}(\mu^{[L]}/\nu, \underline{\lambda}^{[L]}/\nu, \overline{\lambda}^{[L]}/\nu) = p_{\mathcal{C}}^*$. This shows that strong duality holds, and that the dual optimum is attained, at least in the case when $\nu > 0$.

Now we consider the case $\nu = 0$. From (24), we conclude that for any $(x^{[L+1]}, z^{[L]}) \in \mathcal{D}$,

$$\begin{aligned}
\sum_{l=0}^{L-1} \mu^{(l)\top}(z^{(l)} - \mathbf{W}^{(l)} x^{(l)} - b^{(l)}) &+ \sum_{l=0}^{L-1} \underline{\lambda}^{(l)\top}(\underline{\sigma}^{(l)}(z^{(l)}) - x^{(l+1)}) \\
&+ \sum_{l=0}^{L-1} \overline{\lambda}^{(l)\top}(x^{(l+1)} - \overline{\sigma}^{(l)}(z^{(l)})) \geq \alpha \geq 0.
\end{aligned} \tag{25}$$

Taking any feasible point of problem $(\mathcal{C})$, i.e., $(x^{[L+1]}, z^{[L]}) \in \mathcal{S}_{\mathcal{C}}$ and combining with $\underline{\lambda}^{[L]} \geq 0, \overline{\lambda}^{[L]} \geq 0$, we know that the left-hand-side of (25) is non-positive, and thus $\alpha = 0$. Then from (23), we conclude that for any $t \in \mathbb{R}$

$$\|(\underline{u}^{[L]}, \overline{u}^{[L]}, v^{[L]})\|_2 < \frac{p_{\mathcal{C}}^* - t}{C_{\mathcal{C}}} \Rightarrow \underline{\lambda}^{[L]\top} \underline{u}^{[L]} + \underline{\lambda}^{[L]\top} \overline{u}^{[L]} + \mu^{[L]\top} v^{[L]} \leq 0,$$

which can only be possible when $(\underline{\lambda}^{[L]}, \overline{\lambda}^{[L]}, \mu^{[L]}) = 0$. Combined with $\nu = 0$, this contradicts with $(\underline{\lambda}^{[L]}, \overline{\lambda}^{[L]}, \mu^{[L]}, \nu) \neq 0$, and thus $\nu$ cannot be 0. $\qquad \square$

### E.1 Cases where the Slater's condition fails but strong duality holds true by Theorem 4.1

We emphasize that Theorem 4.1 guarantees the strong duality for any pre-specified activation bounds $[\underline{z}^{(l)}, \overline{z}^{(l)}]$ that can be either loose or tight, and for any $\underline{\sigma}^{(l)}$ and $\overline{\sigma}^{(l)}$ that have a finite Lipschitz constant in the domain $[\underline{z}^{(l)}, \overline{z}^{(l)}]$. There are several important cases when Slater's condition does not hold but strong duality holds true by Theorem 4.1.

The first typical scenario is when the pre-specified activation bounds $[\underline{z}^{(l)}, \overline{z}^{(l)}]$ is loose and all the feasible activations $z^{(l)}$ are on the boundary. Let's consider a simple one-layer neural network:

$$\begin{aligned}
x^{(0)} &\in \mathcal{S}_{in}(x^{\text{nom}}), \\
z^{(0)} &= \mathbf{W}^{(0)} x^{(0)} + b^{(0)}, \quad z^{(0)} \in [\underline{z}^{(0)}, \overline{z}^{(0)}], \\
\underline{ReLU}(z^{(0)}) &\leq x^{(1)} \leq \overline{ReLU}(z^{(0)}).
\end{aligned}$$

Suppose that $\mathbf{W}^{(0)} = 0$, $b^{(0)} = -1$, $\underline{z}^{(0)} = -1$ and $\overline{z}^{(0)} = 1$. Then $z^{(0)}$ can only be -1, and $\underline{ReLU}(z^0) = \overline{ReLU}(z^0) = 0$, and thus there does not exist $x^{(1)}$ such that $\underline{ReLU}(z^{(0)}) < x^{(1)} < \overline{ReLU}(z^{(0)})$. In general, orthogonality between $x^{(l)}$ and span($\mathbf{W}^{(l)}$) easily leads to degeneracy of $z^{(l)}$, which can result in the failure of the Slater's condition.

The second typical scenario is when the pre-specified activation bounds in later layers, e.g., $z^{(1)} \in [\underline{z}^{(1)}, \overline{z}^{(1)}]$, forces all feasible points in previous layers, e.g., $x^{(1)}$, to be on the boundary. This degenerate case may occur when one takes the branch-and-bound strategy to split unstable neurons. Let's consider a simple two-layer neural network:

$$\begin{aligned}
&x^{(0)} \in \mathcal{S}_{in}(x^{\mathrm{nom}}) := [-1, 1], \\
&z^{(0)} = x^{(0)}, \quad z^{(0)} \in [-1, 1], \\
&\underline{ReLU}(z^{(0)}) \leq x^{(1)} \leq \overline{ReLU}(z^{(0)}), \\
&z^{(1)} = x^{(1)} - 1, \quad z^{(1)} \in [0, 1], \\
&\underline{ReLU}(z^{(1)}) \leq x^{(2)} \leq \overline{ReLU}(z^{(1)}).
\end{aligned}$$

Due to the pre-specified bound $z^{(1)} \in [0, 1]$, $x^{(1)}$ can only take value 1, which is on the boundary of the nonlinear constraint $\underline{ReLU}(z^{(0)}) \leq x^{(1)} \leq \overline{ReLU}(z^{(0)})$. This leads to failure of the Slater's condition.

After all, there are many edge cases that the Slater's condition does not cover to prove Theorem 4.1. Therefore, we would like to take a novel approach, utilizing the Lipschitz continuity of problem ($\mathcal{C}$), to prove the strong duality without the Slater's condition.

## E.2 Proof of Lemma E.1

Although the proof seems to be long, it is an elementary perturbation analysis for problem ($\mathcal{C}$). We write down every detail so that one can easily check its correctness.

*Proof of Lemma E.1.* When problem (20) is infeasible, i.e., $\widetilde{X}^{(L)} = \emptyset$, $\widetilde{p}_{\mathcal{C}}^* = +\infty$ and (21) naturally holds true. In the following, we prove (21) when problem (20) is feasible.

In this case, we define $X^{(0)} = \widetilde{X}^{(0)} = \mathcal{S}_{in}(x^{\mathrm{nom}})$, $Z^{(l)}$, $\widetilde{Z}^{(l)}$, $X^{(l)}$ and $\widetilde{X}^{(l)}$ recursively as follows:

$$\begin{aligned}
Z^{(l)} &= \{\mathbf{W}^{(l)}x^{(l)} + b^{(l)} : x^{(l)} \in X^{(l)}\} \cap [\underline{z}^{(l)}, \overline{z}^{(l)}], \\
\widetilde{Z}^{(l)} &= \{\mathbf{W}^{(l)}\widetilde{x}^{(l)} + b^{(l)} + v^{(l)} : \widetilde{x}^{(l)} \in \widetilde{X}^{(l)}\} \cap [\underline{z}^{(l)}, \overline{z}^{(l)}], \\
X^{(l+1)} &= \{x^{(l+1)} : \underline{\sigma}^{(l)}(z^{(l)}) \leq x^{(l+1)} \leq \overline{\sigma}^{(l)}(z^{(l)}), z^{(l)} \in Z^{(l)}\}, \\
\widetilde{X}^{(l+1)} &= \{\widetilde{x}^{(l+1)} : \underline{\sigma}^{(l)}(\widetilde{z}^{(l)}) - \underline{u}^{(l)} \leq \widetilde{x}^{(l+1)} \leq \overline{\sigma}^{(l)}(\widetilde{z}^{(l)}) + \overline{u}^{(l)}, \widetilde{z}^{(l)} \in \widetilde{Z}^{(l)}\}.
\end{aligned}$$

Intuitively, $Z^{(l)}$, $\widetilde{Z}^{(l)}$, $X^{(l)}$ and $\widetilde{X}^{(l)}$ are the set of activations that are achievable by the original problem ($\mathcal{C}$) and the perturbed problem (20) given $x^{(0)} \in \mathcal{S}_{in}(x^{\mathrm{nom}})$ and $z^{(l)} \in [\underline{z}^{(l)}, \overline{z}^{(l)}]$. Since both problems are feasible, all the sets above are non-empty.

In the first step, we prove that for every $l \in [L+1]$, there exist positive constants $C_x^{(l)}$ and $C_z^{(l)}$ such that

$$\sup_{\widetilde{x}^{(l)} \in \widetilde{X}^{(l)}} \mathrm{dist}(\widetilde{x}^{(l)}, X^{(l)}) \leq C_x^{(l)} \|(\underline{u}^{[L]}, \overline{u}^{[L]}, v^{[l]})\|_2, \tag{26}$$

$$\sup_{\widetilde{z}^{(l)} \in \widetilde{Z}^{(l)}} \mathrm{dist}(\widetilde{z}^{(l)}, Z^{(l)}) \leq C_z^{(l)} \|(\underline{u}^{[L]}, \overline{u}^{[L]}, v^{[l+1]})\|_2, \tag{27}$$

where $\mathrm{dist}(s, \mathcal{S}) := \inf_{s' \in \mathcal{S}} \|s - s'\|_2$. This means that the perturbation in the achievable activations are "smooth".

Since $X^{(l)} = \widetilde{X}^{(l)} = \mathcal{S}_{in}(x^{\mathrm{nom}})$, we have that (26) holds true for $l = 0$ with $C_x^{(0)} = 0$. In the following, we use mathematical induction to prove (27) for $0 \leq l \leq L-1$ and (26) for $1 \leq l \leq L$.

First, suppose $\mathrm{dist}(\widetilde{x}^{(l)}, X^{(l)}) \leq C_x^{(l)} \|(\underline{u}^{[L]}, \overline{u}^{[L]}, v^{[l]})\|_2$ holds true for any $\widetilde{x}^{(l)} \in \widetilde{X}^{(l)}$. Then for any $\widetilde{z}^{(l)} = \mathbf{W}^{(l)} \widetilde{x}^{(l)} + b^{(l)} + v^{(l)} \in \widetilde{Z}^{(l)}$, we have

$$\mathrm{dist}(\widetilde{z}^{(l)}, Z^{(l)}) := \inf_{z^{(l)} \in Z^{(l)}} \|\widetilde{z}^{(l)} - z^{(l)}\| \leq \inf_{x^{(l)} \in X^{(l)}} \|\mathbf{W}^{(l)}(\widetilde{x}^{(l)} - x^{(l)}) + v^{(l)}\|$$

$$\leq \inf_{x^{(l)} \in X^{(l)}} \|\mathbf{W}^{(l)}\| \|\widetilde{x}^{(l)} - x^{(l)}\| + \|v^{(l)}\| = \|\mathbf{W}^{(l)}\| \mathrm{dist}(\widetilde{x}^{(l)}, X^{(l)}) + \|v^{(l)}\|$$

$$\leq \|\mathbf{W}^{(l)}\| C_x^{(l)} \|(\underline{u}^{[L]}, \overline{u}^{[L]}, v^{[l]})\| + \|v^{(l)}\| \leq \left( (C_x^{(l)})^2 \|\mathbf{W}^{(l)}\|^2 + 1 \right)^2 \|(\underline{u}^{[L]}, \overline{u}^{[L]}, v^{[l+1]})\|.$$

Therefore, (27) holds true with $C_z^{(l)} = \left( (C_x^{(l)})^2 \|\mathbf{W}^{(l)}\|^2 + 1 \right)^2$.

Then by definition, for any $\widetilde{x}^{(l+1)} \in \widetilde{X}^{(l+1)}$, there exists $\widetilde{z}^{(l)} \in \widetilde{Z}^{(l)}$ such that

$$\underline{\sigma}^{(l)}(\widetilde{z}^{(l)}) - \underline{u}^{(l)} \leq \widetilde{x}^{(l+1)} \leq \overline{\sigma}^{(l)}(\widetilde{z}^{(l)}) + \overline{u}^{(l)}.$$

By the induction assumption, there exists $z^{(l)} \in Z^{(l)}$ such that

$$\mathrm{dist}(\widetilde{z}^{(l)}, z^{(l)}) \leq C_z^{(l)} \|(\underline{u}^{[L]}, \overline{u}^{[L]}, v^{[l+1]})\|_2.$$

Thus, we have

$$\mathrm{dist}(\widetilde{x}^{(l+1)}, X^{(l+1)}) = \inf_{x^{(l+1)} \in X^{(l+1)}} \|\widetilde{x}^{(l+1)} - x^{(l+1)}\|$$

$$\leq \inf\{\|\widetilde{x}^{(l+1)} - x^{(l+1)}\| : \underline{\sigma}^{(l)}(z^{(l)}) \leq x^{(l+1)} \leq \overline{\sigma}^{(l)}(z^{(l)})\}.$$

We re-parametrize $\widetilde{x}^{(l+1)}$ and $x^{(l+1)}$ as

$$\widetilde{x}^{(l+1)} = \underline{\sigma}^{(l)}(\widetilde{z}^{(l)}) + \widetilde{t}, \quad x^{(l+1)} = \underline{\sigma}^{(l)}(z^{(l)}) + t,$$

where

$$-\underline{u}^{(l)} \leq \widetilde{t} \leq \Delta\sigma^{(l)}(\widetilde{z}^{(l)}) + \overline{u}^{(l)}, \quad 0 \leq t \leq \Delta\sigma^{(l)}(z^{(l)}),$$
$$\Delta\sigma^{(l)}(\widetilde{z}^{(l)}) = \overline{\sigma}^{(l)}(z^{(l)}) - \underline{\sigma}^{(l)}(z^{(l)}).$$

It is easy to prove that if $\underline{\sigma}^{(l)}$ and $\overline{\sigma}^{(l)}$ have Lipschitz constant $\underline{L}^{(l)}$ and $\overline{L}^{(l)}$ respectively, $\Delta\sigma^{(l)}$ has a Lipschitz constant $\underline{L}^{(l)} + \overline{L}^{(l)}$. Then we have

$$\mathrm{dist}(\widetilde{x}^{(l+1)}, X^{(l+1)}) \leq \|\underline{\sigma}^{(l)}(\widetilde{z}^{(l)}) - \underline{\sigma}^{(l)}(z^{(l)})\| + \inf_{t \in [0, \Delta\sigma^{(l)}(z^{(l)})]} \|\widetilde{t} - t\|$$

$$\leq \underline{L}^{(l)} \|\widetilde{z}^{(l)} - z^{(l)}\| + \left( \sum_k \inf_{t_k \in [0, \Delta\sigma_k^{(l)}(z^{(l)})]} |\widetilde{t}_k - t_k|^2 \right)^{1/2}.$$

We have the entry-wise bound for $\widetilde{t} - t$:

$$\inf_{t_k \in [0, \Delta\sigma_k^{(l)}(z^{(l)})]} |\widetilde{t}_k - t_k|^2 \leq \max(|\underline{u}_k^{(l)}|^2, |\Delta\sigma_k^{(l)}(\widetilde{z}^{(l)}) - \Delta\sigma_k^{(l)}(z^{(l)}) + \overline{u}^{(l)}|^2)$$

$$\leq 2 \left( |\Delta\sigma_k^{(l)}(\widetilde{z}^{(l)}) - \Delta\sigma_k^{(l)}(z^{(l)})|^2 + |\underline{u}_k^{(l)}|^2 + |\overline{u}_k^{(l)}|^2 \right)$$

Therefore, we get

$$\inf_{t \in [0, \Delta\sigma^{(l)}(z^{(l)})]} \|\widetilde{t} - t\| \leq \sqrt{2} \left( \|\Delta\sigma^{(l)}(\widetilde{z}^{(l)}) - \Delta\sigma^{(l)}(z^{(l)})\|^2 + \|\underline{u}_k^{(l)}\|^2 + \|\overline{u}^{(l)}\|^2 \right)^{1/2}$$

$$\leq \sqrt{2} \left( (\underline{L}^{(l)} + \overline{L}^{(l)})^2 \|\widetilde{z}^{(l)} - z^{(l)}\|^2 + \|\underline{u}_k^{(l)}\|^2 + \|\overline{u}^{(l)}\|^2 \right)^{1/2}$$

$$\leq \sqrt{2(\underline{L}^{(l)} + \overline{L}^{(l)})^2 (C_z^{(l)})^2 + 2} \, \|(\underline{u}^{[l+1]}, \overline{u}^{[l+1]}, v^{[l+1]})\|_2$$

Similarly, we have $\underline{L}^{(l)} \|\widetilde{z}^{(l)} - z^{(l)}\| \leq \underline{L}^{(l)} C_z^{(l)} \|(\underline{u}^{[l+1]}, \overline{u}^{[l+1]}, v^{[l+1]})\|_2$. Therefore, we obtain

$$\mathrm{dist}(\widetilde{x}^{(l+1)}, X^{(l+1)}) \leq C_x^{(l+1)} \|(\underline{u}^{[l+1]}, \overline{u}^{[l+1]}, v^{[l+1]})\|_2,$$

where $C_x^{(l+1)} = \underline{L}^{(l)} C_z^{(l)} + \sqrt{2(\underline{L}^{(l)} + \overline{L}^{(l)})^2 (C_z^{(l)})^2 + 2}$.

Then by mathematical induction, we proved that (27) for $0 \le l \le L - 1$ and (26) for $1 \le l \le L$.

In the second step, we prove (21). Thanks to (26) with $l = L$, we have for any $\widetilde{x}^{(L)} \in \widetilde{X}^{(L)}$, there exists $x^{(L)} \in X^{(L)}$ such that

$$\text{dist}(\widetilde{x}^{(L)}, x^{(L)}) \le C_x^{(L)} \|(\underline{u}^{[L]}, \overline{u}^{[L]}, v^{[L]})\|_2.$$

Then we obtain

$$p_{\mathcal{C}}^* - (c^\top \widetilde{x}^{(L)} + c_0) \le c^\top (x^{(L)} - \widetilde{x}^{(L)}) \le \|c\| \|\widetilde{x}^{(L)} - x^{(L)}\|$$
$$\le C_x^{(L)} \|c\| \|(\underline{u}^{[L]}, \overline{u}^{[L]}, v^{[L]})\|_2.$$

Taking the infimum over $\widetilde{x}^{(l)} \in \widetilde{X}^{(l)}$, we have proved (21) with $C_{\mathcal{C}} = C_x^{(L)} \|c\|$. $\qquad \square$

# F   Equivalence of the optimal layer-wise dual relaxations: $d_{\mathcal{C}_{\text{opt}}}^* = d_{\mathcal{O}}^*$

**Lemma F.1.** *Suppose the activation function $\sigma : [\underline{z}, \overline{z}] \to \mathbb{R}$ is bounded from above and below and that $\underline{\sigma}(z) \le \sigma(z) \le \overline{\sigma}(z)$ for all $z \in [\underline{z}, \overline{z}]$. Define*

$$f_{\mathcal{O}}(\mu, \lambda) := \inf_{z \in [\underline{z}, \overline{z}]} \mu z - \lambda \sigma(z), \tag{28}$$

$$f_{\mathcal{C}}(\mu, \underline{\lambda}, \overline{\lambda}) := \inf_{z \in [\underline{z}, \overline{z}]} \mu z + \underline{\lambda} \underline{\sigma}(z) - \overline{\lambda} \overline{\sigma}(z). \tag{29}$$

*For any $\mu$, $\underline{\lambda} \ge 0$ and $\overline{\lambda} \ge 0$, we have*

$$f_{\mathcal{C}}(\mu, \underline{\lambda}, \overline{\lambda}) \le f_{\mathcal{C}}(\mu, -(\overline{\lambda} - \underline{\lambda})_-, (\overline{\lambda} - \underline{\lambda})_+), \tag{30}$$

*where $\lambda_+ = \max(\lambda, 0)$ and $\lambda_- = \min(\lambda, 0)$.*

*When $\underline{\sigma}_{opt}$ and $\overline{\sigma}_{opt}$ are the optimal convex relaxations defined in (12), we write $f_{\mathcal{C}}$ as $f_{\mathcal{C}_{opt}}$. In this case, we have that for any $\mu$ and $\lambda$*

$$f_{\mathcal{C}_{opt}}(\mu, -\lambda_-, \lambda_+) = f_{\mathcal{O}}(\mu, \lambda). \tag{31}$$

*Proof.* First let's prove (30). For $\underline{\lambda} \ge \overline{\lambda} \ge 0$, we have

$$f_{\mathcal{C}}(\mu, -(\overline{\lambda} - \underline{\lambda})_-, (\overline{\lambda} - \underline{\lambda})_+) = f_{\mathcal{C}}(\mu, \underline{\lambda} - \overline{\lambda}, 0)$$

and

$$f_{\mathcal{C}}(\mu, \underline{\lambda}, \overline{\lambda}) = \inf_{z \in [\underline{z}, \overline{z}]} \mu z + \underline{\lambda} \underline{\sigma}(z) - \overline{\lambda} \overline{\sigma}(z) = \inf_{z \in [\underline{z}, \overline{z}]} \mu z + (\underline{\lambda} - \overline{\lambda}) \underline{\sigma}(z) - \overline{\lambda}(\overline{\sigma}(z) - \underline{\sigma}(z))$$

$$\overset{(i)}{\le} \sup_{z \in [\underline{z}, \overline{z}]} \mu z - (\underline{\lambda} - \overline{\lambda}) \underline{\sigma}(z) = f_{\mathcal{C}}(\mu, \underline{\lambda} - \overline{\lambda}, 0),$$

where we use $\overline{\lambda}(\overline{\sigma}(z) - \underline{\sigma}(z)) \ge 0$ in (i). Similarly for $\overline{\lambda} \ge \underline{\lambda} \ge 0$, we have

$$f_{\mathcal{C}}(\mu, -(\overline{\lambda} - \underline{\lambda})_-, (\overline{\lambda} - \underline{\lambda})_+) = f_{\mathcal{C}}(\mu, 0, \overline{\lambda} - \underline{\lambda})$$

and

$$f_{\mathcal{C}}(\mu, \underline{\lambda}, \overline{\lambda}) = \inf_{z \in [\underline{z}, \overline{z}]} \mu z + \underline{\lambda} \underline{\sigma}(z) - \overline{\lambda} \overline{\sigma}(z) = \sup_{z \in [\underline{z}, \overline{z}]} \mu z - (\overline{\lambda} - \underline{\lambda}) \overline{\sigma}(z) - \underline{\lambda}(\overline{\sigma}(z) - \underline{\sigma}(z))$$

$$\overset{(i)}{\le} \sup_{z \in [\underline{z}, \overline{z}]} \mu z - (\overline{\lambda} - \underline{\lambda}) \overline{\sigma}(z) = f_{\mathcal{C}}(\mu, 0, \overline{\lambda} - \underline{\lambda}),$$

where we use $\underline{\lambda}(\overline{\sigma}(z) - \underline{\sigma}(z)) \ge 0$ in (i).

Then let's prove (31). For $\lambda < 0$ ($\lambda_+ = 0$ and $\lambda_- = \lambda$), by definition we have

$$f_{\mathcal{C}_{\text{opt}}}(\mu, -\lambda, 0) = \inf_{z \in [\underline{z}, \overline{z}]} \mu z - \lambda \underline{\sigma}_{\text{opt}}(z) = \lambda \sup_{z \in [\underline{z}, \overline{z}]} \frac{\mu}{\lambda} z - \underline{\sigma}_{\text{opt}}(z)$$

$$\overset{(i)}{=:} \lambda \left(\underline{\sigma}_{\text{opt}}\right)^* (\mu/\lambda) \overset{(ii)}{=} \lambda \left(\sigma\right)^* (\mu/\lambda) \overset{(iii)}{:=} \inf_{z \in [\underline{z}, \overline{z}]} \mu z - \lambda \sigma(z) = f_{\mathcal{O}}(\mu, \lambda),$$

where we use the definition of convex conjugate in (i) and (iii) and the Fenchel-Moreau theorem (Theorem 12.2 in Rockafellar [2015]) in (ii). For $\lambda = 0$, it is obvious. Similarly, for $\lambda > 0$ ($\lambda_+ = \lambda$ and $\lambda_- = 0$), by definition we have

$$f_{\mathcal{C}_{\mathrm{opt}}}(\mu, 0, \lambda) = \inf_{z \in [\underline{z}, \overline{z}]} \mu z - \lambda \overline{\sigma}_{\mathrm{opt}}(z) = -\lambda \sup_{z \in [\underline{z}, \overline{z}]} -\frac{\mu}{\lambda} z - (-\overline{\sigma}_{\mathrm{opt}})(z)$$

$$\overset{(i)}{=:} -\lambda \left(-\overline{\sigma}_{\mathrm{opt}}\right)^* \left(-\mu/\lambda\right) \overset{(ii)}{=} -\lambda \left(-\sigma\right)^* \left(-\mu/\lambda\right) \overset{(iii)}{:=} \inf_{z \in [\underline{z}, \overline{z}]} \mu z - \lambda \sigma(z) = f_{\mathcal{O}}(\mu, \lambda),$$

where we use the definition of convex conjugate in (i) and (iii) and the Fenchel-Moreau theorem in (ii), again. $\qquad \square$

*Proof of Theorem 4.2.* In the first step, we simplify the form of $g_{\mathcal{C}}(\mu^{[L]}, \underline{\lambda}^{[L]}, \overline{\lambda}^{[L]})$. By definition (8), we have

$$g_{\mathcal{C}}(\mu^{[L]}, \underline{\lambda}^{[L]}, \overline{\lambda}^{[L]}) = g^{(0)}(\mu^{(0)}) + \sum_{l=1}^{L-1} g^{(l)}(\mu^{(l)}, \overline{\lambda}^{(l-1)} - \underline{\lambda}^{(l-1)}) + g^{(L)}(c, \overline{\lambda}^{(l-1)} - \underline{\lambda}^{(l-1)})$$

$$+ \sum_{l=0}^{L-1} \left( \widetilde{g}_{\mathcal{C}}^{(l)}(\mu^{(l)}, \underline{\lambda}^{(l)}, \overline{\lambda}^{(l)}) - b^{(l)\top} \mu^{(l)} \right), \tag{32}$$

where

$$g^{(0)}(\mu^{(0)}) = \inf_{x^{(0)} \in \mathcal{S}_{in}(x^{\mathrm{nom}})} \left( -\mathbf{W}^{(0)\top} \mu^{(0)} \right)^\top x^{(0)} \tag{33}$$

$$g^{(l)}(\mu^{(l)}, \lambda^{(l-1)}) = \inf_{x^{(l)}} \left( \lambda^{(l-1)} - \mathbf{W}^{(l)\top} \mu^{(l)} \right)^\top x^{(l)} = \mathbb{1}_{\lambda^{(l-1)} = \mathbf{W}^{(l)\top} \mu^{(l)}}, \tag{34}$$

$$g^{(L)}(c, \lambda^{(L-1)}) = \inf_{x^{(L)}} \left( \lambda^{(L-1)} + c \right)^\top x^{(L)} + c_0 = \mathbb{1}_{\lambda^{(L-1)} = -c} + c_0, \tag{35}$$

and

$$\widetilde{g}_{\mathcal{C}}^{(l)}(\mu^{(l)}, \underline{\lambda}^{(l)}, \overline{\lambda}^{(l)}) = \inf_{\underline{z}^{(l)} \le z^{(l)} \le \overline{z}^{(l)}} \left\{ \mu^{(l)\top} z^{(l)} + \underline{\lambda}^{(l)\top} \underline{\sigma}^{(l)}(z^{(l)}) - \overline{\lambda}^{(l)\top} \overline{\sigma}^{(l)}(z^{(l)}) \right\}. \tag{36}$$

In the second step, for any $\mu^{[L]}$, $\underline{\lambda}^{[L]} \ge 0$ and $\overline{\lambda}^{[L]} \ge 0$, we apply (30) in Lemma F.1 entry-wisely on (36), and obtain

$$\widetilde{g}_{\mathcal{C}}^{(l)}(\mu^{(l)}, \underline{\lambda}^{(l)}, \overline{\lambda}^{(l)}) \le \widetilde{g}_{\mathcal{C}}^{(l)}(\mu^{(l)}, -\lambda_-^{(l)}, \lambda_+^{(l)}),$$

in which $\lambda^{(l)} := \underline{\lambda}^{(l)} - \overline{\lambda}^{(l)}$. After Plugging $\lambda^{[L]} = \lambda_+^{[L]} + \lambda_-^{[L]}$ and $\lambda^{[L]} := \underline{\lambda}^{[L]} - \overline{\lambda}^{[L]}$ into equation (32), we obtain that

$$g_{\mathcal{C}}(\mu^{[L]}, \underline{\lambda}^{[L]}, \overline{\lambda}^{[L]}) \le g_{\mathcal{C}}(\mu^{[L]}, -\lambda_-^{[L]}, \lambda_+^{[L]}). \tag{37}$$

Therefore, the dual problem (9) can be rewritten as an unconstrained optimization problem as

$$d_{\mathcal{C}}^* = \max_{\mu^{[L]}, \lambda^{[L]}} g_{\mathcal{C}}(\mu^{[L]}, -\lambda_-^{[L]}, \lambda_+^{[L]}). \tag{38}$$

In the third step, we simplify $g_{\mathcal{O}}(\mu^{[L]}, \lambda^{[L]})$ based on its definition (10) and obtain

$$g_{\mathcal{O}}(\mu^{[L]}, \lambda^{[L]}) = g^{(0)}(\mu^{(0)}) + \sum_{l=1}^{L-1} g^{(l)}(\mu^{(l)}, \lambda^{(l-1)}) + g^{(L)}(c, \lambda^{(L-1)})$$

$$+ \sum_{l=0}^{L-1} \left( \widetilde{g}_{\mathcal{O}}^{(l)}(\mu^{(l)}, \lambda^{(l)}) - b^{(l)\top} \mu^{(l)} \right) \tag{39}$$

in which

$$\widetilde{g}_{\mathcal{O}}^{(l)}(\mu^{(l)}, \lambda^{(l)}) = \inf_{\underline{z}^{(l)} \le z^{(l)} \le \overline{z}^{(l)}} \mu^{(l)\top} z^{(l)} - \lambda^{(l)\top} \sigma^{(l)}(z^{(l)}). \tag{40}$$

In the forth step, for any $\mu^{[L]}$ and $\lambda^{[L]}$, since all the nonlinear layers are non-interactive, we apply (31) in Lemma F.1 entry-wisely on (36) and (40) and obtain

$$\widetilde{g}_{\mathcal{C}_{\mathrm{opt}}}^{(l)}(\mu^{(l)}, -\lambda_-^{(l)}, \lambda_+^{(l)}) = \widetilde{g}_{\mathcal{O}}^{(l)}(\mu^{(l)}, \lambda^{(l)}).$$

After plugging $\lambda^{[L]} = \lambda_+^{[L]} + \lambda_-^{[L]}$ into (32), we see that the other three terms in $g_{\mathcal{C}_{\mathrm{opt}}}(\mu^{[L]}, -\lambda_-^{[L]}, \lambda_+^{[L]})$ and $g_{\mathcal{O}}(\mu^{[L]}, \lambda^{[L]})$ are the same. Therefore, we have proved that for any $\mu^{[L]}$ and $\lambda^{[L]}$, we have

$$g_{\mathcal{C}_{\mathrm{opt}}}(\mu^{[L]}, -\lambda_-^{[L]}, \lambda_+^{[L]}) = g_{\mathcal{O}}(\mu^{[L]}, \lambda^{[L]}) \tag{41}$$

Finally, combining (11), (38) and (41), we obtain $d_{\mathcal{C}_{\mathrm{opt}}}^* = d_{\mathcal{O}}^*$. $\square$

# G   A greedy algorithm to solve the dual problems

## G.1   Some useful results to simplify the dual problems

We provide the following useful results when solving (9) and (11). First, the dual problem (9) can be rewritten as an unconstrained optimization problem inspired by (38). We define a two-argument function, reusing the name $g_{\mathcal{C}}$, as

$$g_{\mathcal{C}}(\mu^{[L]}, \lambda^{[L]}) := g_{\mathcal{C}}(\mu^{[L]}, -\lambda_-^{[L]}), \lambda_+^{[L]}).$$

Then we have the following useful results.

**Proposition G.1.** *Denote $\lambda_+ = \max(\lambda, 0)$ and $\lambda_- = \min(\lambda, 0)$.*

1. *For dual of the convex relaxed problem ($\mathcal{C}$) defined in (9), we have*

$$d_{\mathcal{C}}^* = \max_{\mu^{[L]}, \lambda^{[L]}} \left\{ g_{\mathcal{C}}(\mu^{[L]}, \lambda^{[L]}) := c_0 + g^{(0)}(\mu^{(0)}) + \sum_{l=0}^{L-1} \left( \widetilde{g}_{\mathcal{C}}^{(l)}(\mu^{(l)}, \lambda^{(l)}) - b^{(l)\top}\mu^{(l)} \right) \right\}, \tag{42}$$

*where*

$$\lambda^{(L-1)} = -c, \quad \lambda^{(l)} = \mathbf{W}^{(l+1)\top}\mu^{(l+1)} \quad \forall l \in [L-1], \tag{43}$$

$$g^{(0)}(\mu^{(0)}) = \inf_{x^{(0)} \in \mathcal{S}_{in}(x^{nom})} \left( -\mathbf{W}^{(0)\top}\mu^{(0)} \right)^\top x^{(0)}, \tag{44}$$

*and*

$$\widetilde{g}_{\mathcal{C}}^{(l)}(\mu^{(l)}, \lambda^{(l)}) = \inf_{\underline{z}^{(l)} \leq z^{(l)} \leq \overline{z}^{(l)}} \left\{ \mu^{(l)\top}z^{(l)} - \lambda_-^{(l)\top}\underline{\sigma}^{(l)}(z^{(l)}) - \lambda_+^{(l)\top}\overline{\sigma}^{(l)}(z^{(l)}) \right\}.$$

2. *For the dual of the original nonconvex problem ($\mathcal{O}$) defined in (11), we have*

$$d_{\mathcal{O}}^* := \max_{\mu^{[L]}, \lambda^{[L]}} \left\{ g_{\mathcal{O}}(\mu^{[L]}, \lambda^{[L]}) = c_0 + g^{(0)}(\mu^{(0)}) + \sum_{l=0}^{L-1} \left( \widetilde{g}_{\mathcal{O}}^{(l)}(\mu^{(l)}, \lambda^{(l)}) - b^{(l)\top}\mu^{(l)} \right) \right\}, \tag{45}$$

*where (43) still holds true and*

$$\widetilde{g}_{\mathcal{O}}^{(l)}(\mu^{(l)}, \lambda^{(l)}) = \inf_{\underline{z}^{(l)} \leq z^{(l)} \leq \overline{z}^{(l)}} \mu^{(l)\top}z^{(l)} - \lambda^{(l)\top}\sigma^{(l)}(z^{(l)}).$$

3. *Suppose that a nonlinear neuron $x_j^{(l+1)} = \sigma^{(l)}(z_{I_j}^{(l)})$ is effectively linear within the input domain $\mathcal{S}_{in}(x^{nom})$, i.e., there exists a linear relation $x_j^{(l+1)} = V_j^{(l)} z_{I_j}^{(l)} + d_j^{(l)}$ for all $x^{(0)} \in \mathcal{S}_{in}(x^{nom})$, then we can simplify the convex relaxed problem ($\mathcal{C}$) by setting*

$$\underline{\sigma}_i^{(l)}(z^{(l)}) = \overline{\sigma}_i^{(l)}(z^{(l)}) = V_j^{(l)} z_{I_j}^{(l)} + d_j^{(l)},$$

*or simplify the original nonconvex problem ($\mathcal{O}$) by setting*

$$\sigma_i^{(l)}(z^{(l)}) = V_j^{(l)} z_{I_j}^{(l)} + d_j^{(l)}.$$

*If this neuron does not interact with other neurons in the same layer, i.e., $z_{I_j}^{(l)}$ is not the input of $x_k^{(l+1)}$ for any $k \neq j$. Then for any optimal point for both dual problems, we have*

$$\mu_{I_j}^{(l)} = V_j^{(l)\top}\lambda_j^{(l)}. \tag{46}$$

Similar results have been obtained in several previous works [Wong and Kolter, 2018, Dvijotham et al., 2018b, Wong et al., 2018, Qin et al., 2019].

*Proof.*

1. (42) is a straightforward rewriting of (32) with (34), (35), (33) and (36).

2. (45) is a straightforward rewriting of (39) with (34), (35), (33) and (40).

3. This can be proved with the same treatment of linear layers in the two items above.

$\square$

## G.2 Greedily solving the dual with linear bounds

Suppose the relaxed bounds $\underline{\sigma}$ and $\overline{\sigma}$ are linear, i.e.,

$$\underline{\sigma}^{(l)}(z^{(l)}) := \underline{a}^{(l)} z^{(l)} + \underline{b}^{(l)}, \quad \overline{\sigma}^{(l)}(z^{(l)}) := \overline{a}^{(l)} z^{(l)} + \overline{b}^{(l)}.$$

In this case, in the dual problem (42) we have

$$d_{\mathcal{C}}^* = \max_{\mu^{[L]}, \lambda^{[L]}} \left\{ g_{\mathcal{C}}(\mu^{[L]}, \lambda^{[L]}) := c_0 + g^{(0)}(\mu^{(0)}) + \sum_{l=0}^{L-1} \left( \widetilde{g}_{\mathcal{C}}^{(l)}(\mu^{(l)}, \lambda^{(l)}) - b^{(l)\top}\mu^{(l)} \right) \right\},$$

where

$$\widetilde{g}_{\mathcal{C}}^{(l)}(\mu^{(l)}, \lambda^{(l)}) = \inf_{\underline{z}^{(l)} \le z^{(l)} \le \overline{z}^{(l)}} \left\{ \left( \mu^{(l)} - \lambda_+^{(l)}\overline{a}^{(l)} - \lambda_-^{(l)}\underline{a}^{(l)} \right) z^{(l)} + \left( \lambda_+^{(l)}\overline{b}^{(l)} - \lambda_-^{(l)}\underline{b}^{(l)} \right) \right\}.$$

In the following, we propose a dual greedy algorithm to *greedily* (approximately) solve the dual problem (9) and/or its simplified version (42). Let $\lambda^{[L]}$ be determined by (43) and $\mu^{[L]}$, for stable neurons, be determined by (46). Both of these are optimal. For *unstable* neurons ($\underline{z}_i^{(l)} \le 0 \le \overline{z}_i^{(l)}$), a suboptimal $\mu^{[L]}$ can be obtained by

$$\mu^{(l)} = \arg\max_{\mu^{(l)}} \widetilde{g}_{\mathcal{C}}^{(l)}(\mu^{(l)}, \lambda^{(l)}),$$

which has a closed form solution

$$\mu_i^{(l)} = \overline{a}_i^{(l)} \left( \lambda_i^{(l)} \right)_+ + \underline{a}_i^{(l)} \left( \lambda_i^{(l)} \right)_- .$$

Notice that the above suboptimal solution for unstable neurons and the optimal solution (46) for stable neurons ($\underline{a}^{(l)} = \overline{a}^{(l)}$ and $\underline{b}^{(l)} = \overline{b}^{(l)}$) can be unified in a single formulae.

Finally, we summarize our algorithm to greedily solve the dual problem as

$$\lambda^{(L-1)} = -c, \quad \mu^{(l)} = \overline{a}^{(l)} \left( \lambda^{(l)} \right)_+ + \underline{a}^{(l)} \left( \lambda^{(l)} \right)_- \quad \lambda^{(l)} = \mathbf{W}^{(l+1)\top}\mu^{(l+1)} \quad \forall l \in [L-1], \quad (47)$$

and the corresponding lower bound is

$$g_{\mathcal{C}}(\mu^{[L]}, \lambda^{[L]}) = c_0 + g^{(0)}(\mu^{(0)}) + \sum_{l=0}^{L-1} \left( \overline{b}^{(l)\top} \left( \lambda^{(l)} \right)_+ - \underline{b}^{(l)\top} \left( \lambda^{(l)} \right)_- - b^{(l)\top}\mu^{(l)} \right). \quad (48)$$

We point out that the algorithm above can exactly recover what was proposed in Theorem 1 in Wong and Kolter [2018]. Their $\nu$ is our $\mu$ and their $\widehat{\nu}$ is our $\lambda$.

# H  Which problem to solve in practice?

Thanks to the strong duality, the same lower bound can be achieved from both the primal and the dual problems, and thus we have the freedom to choose the problem to solve. When the relaxed upper and lower bounds, i.e., $\underline{\sigma}^{(l)}$ and $\overline{\sigma}^{(l)}$, are piece-wise linear (e.g. (4) for ReLU networks), both the primal and dual problems are linear programs and can be efficiently solved by existing LP solvers (which is what we use in the coming sections). In other cases, we recommend to solve the dual problem (11) for two reasons. First, the primal relaxed problem ($\mathcal{C}$) is a constrained optimization problem, and its constraints may not have a simple analytic form when $\underline{\sigma}^{(l)}$ and $\overline{\sigma}^{(l)}$ are not piecewise linear; see examples in Fig. 2. On the contrary, the dual problem (11) can be framed as an unconstrained optimization problem and its objective function has a simple analytic form for some common activation functions [Dvijotham et al., 2018b]. Second, the optimization process of (11) can be stopped anytime to give a lower bound of $p_\mathcal{O}^*$, thanks to weak duality, but this is not true of the primal view. Of course, $\underline{\sigma}^{(l)}$ and $\overline{\sigma}^{(l)}$ must be in the form of (6) to achieve the optimal value.

# I  Additional Experimental Details

## I.1  Neural Networks Used

Here is a list of the network architectures that we use in this paper along with their references if applicable.

**MNIST robust error experiment**

- MLP-A: a multilayer perceptron consisting of 1 hidden layer with 500 neurons [Tjeng et al., 2019].
- MLP-B: a multilayer perceptron consisting of 2 hidden layers with 100 neurons each.

**MNIST $\epsilon$-search experiment**

- CNN-SMALL: ConvNet architecture with two convolutional layers with 16 and 32 filters respectively (size (size $4 \times 4$ and stride of 2 in both), followed by two fully-connected layers with 100 and 10 units respectively [Wong et al., 2018].
- CNN-WIDE-K: ConvNet architecture with two convolutional layers of $4 \times k$ and $8 \times k$ filters (size $4 \times 4$ and stride of 2 in both) followed by a $128 \times k$ fully connected layer followed by two fully-connected layers of sizes $128 \times k$ and 10 respectively. The parameter k is used to control the width of the network [Wong et al., 2018].
- CNN-DEEP-K: ConvNet architecture with $k$ convolutional layers with 8 filters followed by $k$ convolutional filters with 16 filters followed by two fully-connected layers of sizes $100 \times k$ and 10 respectively. The parameter $k$ is used to control the depth of the network [Wong et al., 2018].
- MLP-[9]-500: a multilayer perceptron consisting of 9 hidden layer with 500 neurons each.
- MLP-[9]-100: a multilayer perceptron consisting of 9 hidden layer with 100 neurons each.
- MLP-[2]-100: a multilayer perceptron consisting of 2 hidden layer with 100 neurons each.

**CIFAR-10 $\epsilon$-search experiment**

- CNN-SMALL: ConvNet architecture with two convolutional layers with 16 and 32 filters respectively (size (size $4 \times 4$ and stride of 2 in both), followed by two fully-connected layers with 100 and 10 units respectively.
- CNN-WIDE-K: ConvNet architecture with two convolutional layers of $4 \times k$ and $8 \times k$ filters (size $4 \times 4$ and stride of 2 in both) followed by a $128 \times k$ fully connected layer followed by two fully-connected layers of sizes $128 \times k$ and 10 respectively. The parameter $k$ is used to control the width of the network.

## I.2  Training Modes

In this paper, we use only one pre-trained network from the literature, and we train the rest from scratch.

**Pre-trained Networks**

- ADV-MLP-A: this is a multilayer perceptron with 1 hidden layer having 500 units. It is trained using PGD with $l_\infty$ perturbation of $\epsilon = 0.1$, and is used in Tjeng et al. [2019] and Raghunathan et al. [2018a]. It can be found at `https://github.com/vtjeng/MIPVerify_data/tree/master/weights/mnist/RSL18a`.

**Networks Trained from Scratch.**   We train all models in parallel on a GPU-cluster with P100 GPUs.

- All networks in the paper that have the prefix or training mode ADV are trained with PGD using the code available at `https://github.com/locuslab/convex_adversarial/blob/master/examples/mnist.py`.

- All networks in the paper that have the prefix or training mode LPD are trained with the robust training method of Wong et al. [2018] using the code available at `https://github.com/locuslab/convex_adversarial/blob/master/examples/mnist.py`.

- All networks in the paper that have the prefix NOR or training mode NORMAL are trained the regular cross-entropy loss using the code available at `https://github.com/locuslab/convex_adversarial/blob/master/examples/mnist.py`.

- All the CIFAR-10 networks in the paper have the same naming convention as above, but are trained using the code available at `https://github.com/locuslab/convex_adversarial/blob/master/examples/cifar.py`.

## J  Parallel Computation Details

**Why do we need parallel computing to solve LP-ALL?**   The nature of our LP-ALL algorithm requires solving a number of LP that scales with the number of neurons in the network we are verifying. For example, if we want to verify a network with 10k neurons on ten samples the MNIST dataset. We need to solve roughly 10k LPs/sample $\times$ 10 samples $=$ 100k LPs.

The average time for solving an LP varies with the size of the network (see Fig. 5 and 6). It also varies depending on which layer in the network the neuron, for which we are solving the LP, is in (see Fig. 7). Let us say on average the duration for solving one LP is 10 sec on the CPUs we use, which is reasonable for networks that we consider in this paper. *Therefore, for verifying one network, we need around 1 million sec which is roughly 11 days.*

Doing this for all the models in the paper and for more samples would take years. This is why parallelizing the computation was crucial. Therefore we conduct all the experiments on a cluster with **1000 CPU-nodes**. Another key point here was to make sure that the scheduling pipeline on the cluster has **very low latency**, because we need to solve around 100 million jobs in total in the paper, each of which is on the order of seconds. So any latency in the pipeline can cause significant overhead. The details of the scheduling pipeline are beyond the scope of this paper.

**CPU specifications.**   Each CPU-node we used has 2 virtual CPUs with a 2.4 GHz Intel(R) Xeon(R) E5-2673 v3 (Haswell) processor and 7GB of RAM.

**Linear programming (LP) solver used.**   We construct all the LP models in python using CVXPY [Diamond and Boyd, 2016], and the models are solved using an open-source solver, ECOS [Domahidi et al., 2013]. We found this solver to be the fastest among other open-source solvers for our application.

## K  Computational Time for Solving LP-ALL

The solve time of the LP in ($\mathcal{C}$) depends mainly on the size and the training method of a neural network. It also depends on the input-space dimension.

**Dependence on architecture and training mode.**   Fig. 5 and 6 shows the average solve time of the LP in ($\mathcal{C}$) for various networks and training methods that are used in the paper on MNIST and CIFAR-10 datasets, respectively. This averaging is over all the neurons in each network, and over ten samples of each dataset. Note how the solve time increases as the network becomes wider or deeper. This is because the number of decision variables and constraints in the LP increases as the network becomes wider or deeper. Another observation is that, in contrast to MILP [Tjeng et al., 2019], the solve time for robustly trained networks seems to be larger than those which are trained using the regular cross-entropy loss or those which are randomly initialized. This is possibly due to the fact that we are not exploiting the stability of neurons in our implementation of the LP as opposed to what is done in the MILP implementation of Tjeng et al. [2019].

**Dependence on which layer we are solving for.**   Fig. 7 shows the average solve time per neuron per layer of the LP in ($\mathcal{C}$) for each of the networks that are used in the paper on the CIFAR-10 dataset. Notice how the solve time of the LP increases as we go deeper into the network.

Figure 5: Average duration for solving the LPs for each model (averaged over the neurons in the model and over 10 samples of the MNIST dataset.

Figure 6: Average duration for solving the LPs for each model averaged over the neurons in the model and over 10 samples of the CIFAR-10 dataset.

Figure 7: Average duration for solving the LPs per layer for each model averaged over the neurons in the model and over 10 samples of the CIFAR-10 dataset.

# L   Full Results of Certified Bounds on the Minimum Adversarial Distortion Experiment

## L.1   Implementation details

In this experiment, we are interested in searching for the minimum adversarial distortion $\epsilon$, which is the $l_\infty$ radius of largest $l_\infty$ ball in which no adversarial examples can be crafted.

An upper bound on $\epsilon$ can be calculated by using PGD in a binary search setting: given an initial guess of $\epsilon$, PGD can be used to find an adversarial example. If successful, divide $\epsilon$ by 2; else multiply $\epsilon$ by 2; and repeat until the change in $\epsilon$ is below some tolerance ($10^{-5}$ in our case).

Lower bounds on $\epsilon$ are calculated using LP-GREEDY, LP-LAST , or our LP-ALL algorithm in a binary search setting; given an initial guess of $\epsilon$, any of these algorithms can be used to check whether the network is robust within $\epsilon$-perturbation of the input. If robust, multiply $\epsilon$ by 2; else divide $\epsilon$ by 2; and repeat until the change in $\epsilon$ is below a tolerance. The tolerances used in the paper are:

- $\text{tol}(\epsilon_{\text{LP-GREEDY}}) = 10^{-5}$ because LP-GREEDY is computationally very cheap.
- $\text{tol}(\epsilon_{\text{LP-LAST}}) = 5\% \times \epsilon_{\text{LP-GREEDY}}$ because LP-LAST is computationally expensive.
- $\text{tol}(\epsilon_{\text{LP-ALL}}) = 5\% \times \epsilon_{\text{LP-GREEDY}}$ because LP-ALL is computationally expensive.

Since solving LP-ALL is really expensive, we find the $\epsilon$-bounds only for ten samples of the MNIST and CIFAR-10 datasets. In this experiment, both ADV- and LPD-networks are trained with an $l_\infty$ maximum allowed perturbation of 0.1 and $8/255$ on MNIST and CIFAR-10, respectively. The full results are reported in Tables 2 and 3 respectively.

## L.2   Results

Tables 2 and 3 both report, for ten samples of MNIST and CIFAR-10 respectively, for a wide range of networks :

1. The training mode, whether the network is trained using regular CE loss (Normal), using adversarial examples generated by PGD (ADV), or using the robust loss in Wong and Kolter [2018] (LPD).
2. Mean lower bounds on $\epsilon$ found by LP-GREEDY, LP-LAST, and LP-ALL. Note that naturally
$$\epsilon_{\text{LP-GREEDY}} \leq \epsilon_{\text{LP-LAST}} \leq \epsilon_{\text{LP-ALL}}$$
3. A mean upper bound on $\epsilon$ found by PGD.
4. The median percentage gap between PGD and the three LP-relaxed algorithms. The percentage gap is defined as
$$\%\text{gap} = \frac{(\epsilon_{\text{PGD}} - \epsilon_{\text{LP-X}})}{\epsilon_{\text{PGD}}} \times 100.$$
   It is also easy to see that naturally,
$$\%\text{gap}_{\text{LP-GREEDY}} \geq \%\text{gap}_{\text{LP-LAST}} \geq \%\text{gap}_{\text{LP-ALL}}$$

The results of both tables show that for all networks, the certified lower bounds on $\epsilon$ using LP-GREEDY, LP-LAST, or LP-ALL are 1.5 to 5 times smaller than the upper bound found by PGD on MNIST, and 1.5 to 2 times smaller than the upper bound found by PGD on MNIST. This gap can also clearly be observed in Fig. 3 and Fig. 8 for MNIST and CIFAR-10, respectively.

Therefore, the improvement that we get using LP-ALL and LP-LAST over LP-GREEDY is not significant and doesn't close the gap with the PGD upper bound.

# M   Results on Randomly Initialized Networks

In this section, we report additional results for the $\epsilon$-search experiment because they might be of interest as a comparison. The results are reported in Table 4. The results are in accordance to what was discussed in Seciton 6.2 i.e. for all networks and both datasets, the certified lower bounds on $\epsilon$ using LP-GREEDY, LP-LAST, or LP-ALL are 2 to 3 times smaller than the upper bound found by PGD. Furthermore, the improvement that we get using LP-ALL and LP-LAST over LP-GREEDY is not significant and doesn't close the gap with the PGD upper bound.

Table 2: Certified bounds on the minimum adversarial distortion $\epsilon$ for ten random samples from the test set of MNIST.

| NETWORK | TRAINING MODE | MEAN LOWER BOUND ($\times 10^{-3}$) | | | MEAN UPPER BOUND ($\times 10^{-3}$) | MEDIAN PERCENTAGE GAP (%) | | |
|---|---|---|---|---|---|---|---|---|
| | | LP-GREEDY | LP-LAST | LP-ALL | PGD | LP-GREEDY | LP-LAST | LP-ALL |
| CNN-SMALL | NORMAL | 14.98 | 16.29 | 18.87 | 52.70 | 69.12 | 66.03 | 61.40 |
| | ADV | 73.42 | 77.09 | 85.94 | 155.16 | 52.52 | 50.14 | 44.42 |
| | LPD | 153.17 | 160.83 | 160.83 | 226.72 | 29.72 | 26.21 | 26.21 |
| CNN-WIDE-1 | NORMAL | 14.09 | 15.76 | 16.92 | 39.61 | 58.84 | 54.69 | 52.66 |
| | ADV | 81.52 | 86.25 | 91.76 | 142.89 | 43.59 | 40.77 | 37.58 |
| | LPD | 116.72 | 122.55 | 122.55 | 183.66 | 33.90 | 30.59 | 30.59 |
| CNN-WIDE-2 | NORMAL | 13.29 | 14.83 | 16.82 | 43.95 | 68.34 | 64.40 | 60.42 |
| | ADV | 91.50 | 96.08 | 104.02 | 179.86 | 49.83 | 47.32 | 41.98 |
| | LPD | 148.07 | 156.78 | 169.77 | 221.67 | 32.45 | 27.76 | 21.03 |
| CNN-WIDE-4 | NORMAL | 12.84 | 14.37 | 16.45 | 47.23 | 72.23 | 68.06 | 63.06 |
| | ADV | 67.64 | 72.34 | 79.90 | 178.01 | 62.72 | 59.37 | 55.17 |
| | LPD | 142.30 | 149.41 | 155.23 | 217.64 | 34.92 | 31.67 | 29.34 |
| CNN-WIDE-8 | NORMAL | 10.82 | 11.72 | 13.35 | 47.75 | 75.49 | 71.85 | 69.36 |
| | ADV | 62.57 | 67.42 | 77.66 | 181.09 | 64.45 | 62.17 | 55.57 |
| | LPD | N.A | N.A | N.A | N.A | N.A | N.A | N.A |
| CNN-DEEP-1 | NORMAL | 15.21 | 16.78 | 19.58 | 44.79 | 66.50 | 62.04 | 55.44 |
| | ADV | 94.68 | 99.41 | 100.20 | 166.38 | 39.81 | 36.80 | 35.93 |
| | LPD | 136.09 | 142.89 | 142.89 | 184.23 | 22.10 | 18.20 | 18.20 |
| CNN-DEEP-2 | NORMAL | 6.12 | 6.42 | 8.76 | 43.32 | 84.47 | 83.69 | 78.65 |
| | ADV | 102.47 | 107.60 | 112.82 | 185.70 | 39.35 | 36.32 | 36.32 |
| | LPD | N.A | N.A | N.A | N.A | N.A | N.A | N.A |
| MLP-[9]-500 | NORMAL | 12.64 | 13.27 | 16.84 | 45.84 | 74.57 | 73.30 | 63.14 |
| | ADV | 20.77 | 21.99 | 28.50 | 129.45 | 84.60 | 83.83 | 79.05 |
| | LPD | N.A | N.A | N.A | N.A | N.A | N.A | N.A |
| MLP-[9]-100 | NORMAL | 11.35 | 11.92 | 14.23 | 31.37 | 64.13 | 62.34 | 57.03 |
| | ADV | 19.41 | 21.12 | 25.41 | 94.57 | 75.15 | 71.42 | 63.96 |
| | LPD | 68.25 | 71.51 | 73.96 | 103.87 | 29.79 | 26.28 | 26.28 |
| MLP-[2]-100 | NORMAL | 14.19 | 15.11 | 15.83 | 28.14 | 52.66 | 47.82 | 45.56 |
| | ADV | 41.68 | 43.76 | 43.76 | 81.22 | 36.23 | 33.04 | 33.04 |
| | LPD | 81.50 | 85.33 | 85.33 | 118.10 | 25.01 | 21.26 | 21.26 |

Table 3: Certified bounds on the minimum adversarial distortion $\epsilon$ for ten random samples from the test set of CIFAR-10.

| NETWORK | TRAINING MODE | MEAN LOWER BOUND ($\times 10^{-3}$) | | | MEAN UPPER BOUND ($\times 10^{-3}$) | MEDIAN PERCENTAGE GAP (%) | | |
|---|---|---|---|---|---|---|---|---|
| | | LP-GREEDY | LP-LAST | LP-ALL | PGD | LP-GREEDY | LP-LAST | LP-ALL |
| CNN-SMALL | NORMAL | 7.48 | 7.86 | 8.46 | 20.13 | 49.40 | 46.87 | 44.23 |
| | ADV | 24.33 | 26.53 | 27.59 | 37.90 | 34.50 | 24.67 | 24.67 |
| | LPD | 67.34 | 72.27 | 77.84 | 157.01 | 52.94 | 48.13 | 43.13 |
| CNN-WIDE-1 | NORMAL | 6.97 | 7.32 | 7.56 | 14.57 | 43.01 | 40.16 | 39.39 |
| | ADV | 58.52 | 63.26 | 67.84 | 115.47 | 49.83 | 46.63 | 42.15 |
| | LPD | 57.03 | 62.51 | 65.83 | 122.00 | 41.22 | 38.29 | 32.40 |
| CNN-WIDE-2 | NORMAL | 8.27 | 8.86 | 9.46 | 22.16 | 58.66 | 54.53 | 52.46 |
| | ADV | 42.05 | 45.99 | 49.09 | 74.13 | 35.10 | 29.85 | 25.54 |
| | LPD | 73.19 | 81.75 | 87.38 | 157.03 | 47.64 | 39.78 | 39.78 |
| CNN-WIDE-4 | NORMAL | 4.14 | 4.35 | 4.63 | 10.97 | 40.27 | 37.28 | 33.03 |
| | ADV | 29.11 | 32.84 | 35.45 | 71.57 | 50.59 | 44.21 | 43.18 |
| | LPD | 41.62 | 47.17 | 48.51 | 104.49 | 45.19 | 39.67 | 39.67 |

Figure 8: The median percentage gap of minimum adversarial distortion for CIFAR-10, in the same format as Fig. 3. For more details, please refer to Table 3 in Appendix L.2.

Table 4: Certified bounds on the minimum adversarial distortion $\epsilon$ for ten random samples from the test set of MNIST and CIFAR-10 on randomly initialized networks (no training).

| NETWORK | TRAINING MODE | MEAN LOWER BOUND ($\times 10^{-3}$) | | | MEAN UPPER BOUND ($\times 10^{-3}$) | MEDIAN PERCENTAGE GAP (%) | | |
|---|---|---|---|---|---|---|---|---|
| | | LP-GREEDY | LP-LAST | LP-ALL | PGD | LP-GREEDY | LP-LAST | LP-ALL |
| **MNIST** | | | | | | | | |
| CNN-SMALL | RANDOM | 5.79 | 6.08 | 6.25 | 14.86 | 51.37 | 48.94 | 48.94 |
| CNN-WIDE-1 | RANDOM | 10.42 | 10.94 | 11.98 | 33.77 | 67.09 | 65.45 | 62.16 |
| CNN-WIDE-2 | RANDOM | 8.12 | 8.53 | 9.34 | 29.43 | 72.54 | 71.17 | 68.42 |
| CNN-WIDE-4 | RANDOM | 8.68 | 9.12 | 9.99 | 45.26 | 78.65 | 77.59 | 75.45 |
| CNN-DEEP-1 | RANDOM | 11.12 | 11.81 | 12.79 | 42.28 | 72.76 | 71.40 | 68.67 |
| MLP-[2]-100 | RANDOM | 4.69 | 5.16 | 5.25 | 15.71 | 64.85 | 58.53 | 57.83 |
| **CIFAR-10** | | | | | | | | |
| CNN-SMALL | RANDOM | 8.77 | 10.01 | 10.13 | 24.50 | 62.61 | 57.04 | 57.04 |
| CNN-WIDE-1 | RANDOM | 5.61 | 5.89 | 6.09 | 11.33 | 45.27 | 42.53 | 41.46 |
| CNN-WIDE-2 | RANDOM | 2.83 | 3.31 | 3.31 | 6.24 | 50.60 | 46.13 | 46.13 |
| CNN-WIDE-4 | RANDOM | 8.93 | 8.52 | 9.00 | 28.69 | 69.63 | 68.11 | 68.11 |