[Reviews · NeurIPS 2019]

Reviewer 1



The paper studies relaxed verifiers and argue that the LP-based verifiers can all be modeled using their convex-relaxation based framework. It then shows that convex-relaxed based classifiers (both based on primal relaxation and dual relaxation) cannot find tight bounds. This is shown empirically by comparing with PGD-based heuristic and MIP-based exact method. The paper is well written and seems to cite related work properly. In terms of methods, it does not seem to propose anything significant however it does a nice job in relating many of the known verifiers which is important as it then illustrates that the bounds they produce are loose.

Reviewer 2



This paper addresses a timely topic very well and gives us a better understanding on the quality of layer-wise convex relaxations. It is well-written, convincing, and covers the topics discussed in a lot of detail. The discussions included in the paper are interesting and provide a lot of context, which makes the paper readable to someone with a passing interest in the topic. [ Update after author feedback ] In my opinion the authors have sufficiently responded to the concerns of the reviewers.

Reviewer 3



The paper targets the problem of robustness verification of neural networks. This is a very popular and important problem. One of the prominent ways to deal with it is by formulating it as a nonlinear optimization problem and then relaxing its constraints to form a linear program relaxation. These relaxations are not guaranteed to return the optimal value, but they can be solved in polynomial time and provide bounds on the optimal solution. The main contributions of the paper are as follows: 1. Proposing a unified framework that generalizes all known layerwise LP relaxations, and showing their relationship (i.e., which relaxation is tighter). 2. Showing that the performance of all methods within this framework is theoretically limited to the performance of the optimal layerwise relaxation 3. Computing the optimal relaxation for various ReLU networks, comparing it to existing relaxations, and showing that it does not significantly outperform them. The authors conclude that there is a “performance barrier” for layerwise LP relaxations. In general, the paper is well written, and the results seem significant and essential for future work. Additional comments: 1. The figures are in low resolution. Please change to high resolution. 2. Equations 3 - It might be clearer to spread it on more rows. Explaining each constraint in this formulation might help readability. 3. Figure 1: where are the proofs for all the relaxation relationships? Please discuss this or point to the appropriate place in the appendix. I am also missing a discussion on why all these relaxations fall into this framework. ------------------------------------------------------------------------------------------------------- Post-rebuttal ------------------------------------------------------------------------------------------------------- I have read the other reviews and author response. I still feel this is a good submission that will help future work. The additional explanations the authors provided in their response regarding the relationships between the relaxations are important.

[Author Response · NeurIPS 2019]

We thank the reviewers for their time, effort, and helpful feedback. We will make sure to incorporate the reviewers' suggestions in the final version of the paper. We address individual feedback below. The citation keys are the same with the main paper, with one additional reference for this rebuttal.

**Reviewer 1:** *"... a major takeaway I get is that the PGD attack seems to provide an okay approximation of robustness ... I would be interested in seeing how the other verifiers (which do not fall within the convex-relaxed framework) are compared to the PGD attacks."*

Reply: First, we affirm that "the PGD attack seems to provide an okay approximation of robustness compared to the layer-wise convex-relaxed methods" is an empirical observation, based on our empirical results in Table 1 and empirical results in other papers, e.g., [Tjeng et al., 2019, Xiao et al., 2019]. However, these empirical results are only obtained for small-size networks, where obtaining the exact answer is feasible via MILP.

Second, verifiers outside our framework, e.g., [Raghunathan et al., 2018b, Bunel et al., 2018, Singh et al., 2019b], may provide tighter upper bounds as compared to the upper bounds of the relaxed verifiers within our framework. However, these verifiers are significantly more computationally expensive than LP relaxed classifiers (which already take 22 CPU-years to benchmark in our paper), and thus a careful and comprehensive experiment should be designed to benchmark them. It is one important future work to be done in this field, but it is outside the scope of our paper.

After all, PGD is a more **practical** approximation of robustness for networks, compared to convex relaxation based methods. However, for a randomized smoothing classifier (which uses a base network but is not a network itself, see, e.g., Cohen et al., ICML 2019), the certified robustness provided by Cohen et al. is easy to compute and tight, and can **also** serve as a good approximation of robustness for randomized smoothing classifiers. In addition, we emphasize that PGD provides **lower bounds** on the robust error while relaxed-verifiers provide **upper bounds**. We should be cautious when comparing these two different types of bounds.

Please consider raising your score if you like our work.

**Reviewer 2:** Thank you for your positive review!

**Reviewer 3:** *"Please change figures to high resolution."* Reply: Thanks, will do in the revised version.

*"Equations 3 - It might be clearer to spread it on more rows. Explaining each constraint in this formulation might help readability."* Reply: Thanks, will reformat the equation and add explanations and intuitions in the revised version.

*"Proofs and discussions of relaxation relationships in Figure 1"* Reply: The edges labeled with Theorem 4.2 and Collorary 4.3 are our main theoretical contributions. In the following, we provide discussions about other relationships, which we will make clear and provide pointers to places in the appendix in the revised version.

"Optimal layer-wise convex relaxation -> CROWN": trivially holds since CROWN is a greedy algorithm to solve LP relaxations (problem $\mathcal{C}$ plus Eq. (7)), which can be included in the convex relaxation framework (Appendix D).

"CROWN -> Fast-Lin": Zhang et al., NIPS 2018 proposed CROWN as a more general variant of Fast-Lin. In Fast-Lin, the linear relaxation needs to use the same slope for upper and lower bounds; in CROWN, the slope can be different. In other words, in Eq. (7) the $\overline{a}^{(l)} = \underline{a}^{(l)}$ for Fast-Lin but this is not a requirement anymore for CROWN.

"LP-Relaxed Dual -> CROWN" and "LP-Relaxed Dual -> Fast-Lin": CROWN and Fast-Lin uses one linear upper bound and one linear lower bound (Eq. (7)), so problem $\mathcal{C}$ becomes a special case of LP-relaxed problem. Especially, line 183-188 and line 572-576 discussed the relationship between dual-LP, Fast-Lin and CROWN.

"DeepPoly <-> CROWN" and "DeepZ <-> Fast-Lin": A simple comparison of these algorithms would show them to be the same. While different papers use different notations, one can straightforwardly translate between them. In (Singh et al., ICLR 2019) page 9, section 4.2, the author (same author as DeepZ and DeepPoly) commented "We note that DeepZ has *the same* precision as Fast-Lin (Weng et al., ICML 2018) and DeepPoly has *the same* precision as CROWN (Zhang et al., NIPS 2018)." Also in the experiments of DeepZ (Singh et al., NIPS 2018), the lines for DeepZ and Fast-Lin overlap (exactly the same values are obtained). Note that these algorithms have slight differences (for example, DeepPoly and DeepZ consider floating point rounding issues and have a faster implementation), but the basic algorithm computes exactly the same bounds.

"DeepPoly -> DeepZ": Relationship is similar to "CROWN -> FastLin". Line 553-565 in Appendix D briefly discussed the relaxation relationship between these algorithms.

"Optimal layer-wise convex relaxation -> DeepPoly": holds because CROWN and DeepPoly use the same relaxation and algorithm to greedily solve the resulting linear programming problem.

"Fast-Lin <-> Neurify": Fast-Lin and Neurify uses the same relaxation for ReLU neurons (and unlike other works, these two only deal with ReLU activation functions). This can be observed by comparing figure 3 in Neurify paper and Figure 1 in the Fast-Lin paper: the selection of slope $\underline{a}^{(l)}$ and $\overline{a}^{(l)}$ are the same (line 513 in Appendix D).

**Additional references**    Cohen, J.M., Rosenfeld, E. and Kolter, J.Z.. Certified adversarial robustness via randomized smoothing. ICML 2019

[Meta-Review · NeurIPS 2019]

An interesting paper on robustness verification. The paper proposes a general framework for layer-wise LP relaxations and shows which relaxation is tighter. Further it shows that there is a theoretical barrier to layer-wise LP relaxations.